# E2F6 initiates stable epigenetic silencing of germline genes during embryonic development

Thomas Dahlet [1,2,8], Matthias Truss[3,8✉], Ute Frede[3], Hala Al Adhami [1,2], Anaïs F. Bardet [1,2], Michael Dumas[1,2], Judith Vallet[1,2], Johana Chicher [4], Philippe Hammann [4], Sarah Kottnik[3], Peter Hansen[5], Uschi Luz[3], Gonzalo Alvarez [3], Ghislain Auclair[1,2], Jochen Hecht[5,6], Peter N. Robinson [5,7], Christian Hagemeier [3✉] & Michael Weber [1,2✉]

In mouse development, long-term silencing by CpG island DNA methylation is specifically targeted to germline genes; however, the molecular mechanisms of this specificity remain unclear. Here, we demonstrate that the transcription factor E2F6, a member of the polycomb repressive complex 1.6 (PRC1.6), is critical to target and initiate epigenetic silencing at germline genes in early embryogenesis. Genome-wide, E2F6 binds preferentially to CpG islands in embryonic cells. E2F6 cooperates with MGA to silence a subgroup of germline genes in mouse embryonic stem cells and in embryos, a function that critically depends on the E2F6 marked box domain. Inactivation of *E2f6* leads to a failure to deposit CpG island DNA methylation at these genes during implantation. Furthermore, E2F6 is required to initiate epigenetic silencing in early embryonic cells but becomes dispensable for the maintenance in differentiated cells. Our findings elucidate the mechanisms of epigenetic targeting of germline genes and provide a paradigm for how transient repression signals by DNA-binding factors in early embryonic cells are translated into long-term epigenetic silencing during mouse development.

[1] University of Strasbourg, Strasbourg, France. [2] CNRS UMR7242, Biotechnology and Cell Signaling, Illkirch, France. [3] Pediatric Oncology, Labor für Pädiatrische Molekularbiologie, Charité - Universitätsmedizin Berlin, Berlin, Germany. [4] Plateforme protéomique Strasbourg Esplanade, CNRS, University of Strasbourg, Strasbourg, France. [5] Berlin Brandenburg Center for Regenerative Therapies (BCRT), Charité Universitätsmedizin Berlin, Berlin, Germany. [6] Present address: Centre for Genomic Regulation, Barcelona, Spain. [7] Present address: Jackson Laboratory for Genomic Medicine, Farmington, CT, USA. [8] These authors contributed equally: Thomas Dahlet, Matthias Truss. ✉email: matthias.truss@charite.de; christian.hagemeier@charite.de; michael.weber@unistra.fr

Methylation of cytosines in CpG dinucleotides by the DNA methyltransferases (DNMTs) is one of the best-characterized epigenetic control mechanisms for gene transcription. In mice, genome-wide DNA methylation patterns are established during a short wave of de novo methylation concomitant to embryo implantation[1,2]. During this developmental stage, *Dnmt3a* and *Dnmt3b* deposit DNA methylation throughout the genome[3].

In contrast to interspersed CpG dinucleotides, the majority of genomic regions of high CpG density, known as CpG islands (CGIs), are protected against DNA methylation. In post-implantation embryos, <1% of CGIs at transcription start sites (TSSs) have acquired DNA methylation[3]. The mechanisms underlying the protection of CGIs from DNA methylation are still poorly understood. CGIs tether proteins containing CXXC zinc finger DNA-binding domains that recognize unmethylated CpG-containing DNA. Some of these CXXC proteins have been implicated in the protection against DNA methylation[4]. CGIs also harbor high levels of H3K4 trimethylation known to inhibit the activity of DNMT3 methyltransferases and de novo DNA methylation[5–8]. In addition, several studies suggest that binding of activating transcription factors (TFs) at CGIs contributes to protection against DNA methylation[9–13].

Strikingly, promoter CGI methylation in development is almost exclusively observed at germline genes[3]. Suppression of transcription of germline genes in somatic cells depends on DNA methylation, as they are derepressed in DNA methylation-deficient mouse embryos and in response to inhibitors of DNA methylation[3,14,15]. However, the molecular mechanisms leading to DNA methylation, specifically at CGIs of germline genes, are unknown. Existing evidence suggests that the TF E2F6 could play a role in this process. E2F6 is a member of the E2F family of TFs that control cell proliferation and cell fate[16–19]. E2F6 serves as a transcriptional repressor that binds the DNA site 5′-TCCCGC-3′, which is highly conserved in promoter regions of meiosis-specific genes[20]. For the *Tuba3a* gene, E2F6-dependent gene repression in somatic cells involves promoter DNA methylation[21]. Interestingly, somatic cells lacking E2F6 or DNMT3B show overlapping sets of derepressed genes[1,22], further suggesting a link between E2F6 and DNA methylation. Based on experiments in somatic cells, a direct recruitment of DNMT3B by E2F6 has been suggested[22].

In mammalian cells, E2F6 interacts with components of the polycomb repressive complex PRC1.6[23–28]. PRC1.6 is one of the non-canonical PRC1 (ncPRC1) complexes that are recruited to chromatin independently of H3K27 methylation. Targeting of ncPRC1 complexes can involve recognition of unmethylated CGIs through the CXXC domain of KDM2B[29]. The PRC1.6 complex lacks the KDM2B subunit but contains RYBP, PCGF6, L3MBTL2, RING1A/B, CBX3, YAF2, WDR5, and the DNA-binding TFs E2F6/DP1 and MGA/MAX[24,26]. Interestingly MAX, MGA, PCGF6, L3MBTL2, and RYBP have been linked to the repression of germline genes in mouse embryonic stem cells (ESCs)[24,30–34]. In contrast to the other components of PRC1.6, the biological roles and targets of E2F6 have been less explored. Here we investigated the biological functions of E2F6 in the mouse and the hypothesis that E2F6 may play an essential role in initiating long-term epigenetic repression of germline genes during embryogenesis.

## Results

### E2F6 genomic targets and protein partners in mouse ESC cells.

To start investigating the role of E2F6, we performed chromatin immunoprecipitation-sequencing (ChIP-seq) analysis of E2F6 in $E2f6^{+/+}$ and $E2f6^{-/-}$ mouse ESCs established from E2f6-

knockout (KO) and control mice[21] (Supplementary Data 1). This analysis generated 2533 high confidence peaks (Supplementary Data 2). The established E2F6 target gene *Tuba3a*[21] revealed a well-defined peak over its TSS (Fig. 1a). Importantly, all peaks were not detected in control ChIP-seq derived from $E2f6^{-/-}$ ESCs, thereby confirming the specificity of the immunoprecipitation (Fig. 1a and Supplementary Fig. 1a). The majority of E2F6 peaks were located close to TSSs and colocalize with CGIs (Fig. 1b, c). Sequence motif analysis of the E2F6 ChIP-seq peaks revealed that they are enriched for the E2F6-binding site 5′-TCCCGC-3′ (2153/2533 peaks, $p = 2.91e - 129$) and the E-box motif 5′-CACGTG-3′ bound by MAX/MGA (661/2533 peaks, $p = 3.80e - 65$) (Fig. 1d, e and Supplementary Fig. 1b). Gene Ontology (GO) analysis of E2F6-bound gene promoters revealed an enrichment for gene functions associated with cell cycle, DNA replication, response to DNA damage, and meiosis (Supplementary Data 2), which is in line with known functions of E2F6 that binds to genes activated by E2F TFs during the G1/S transition of the cell cycle[17,35].

Next, we compared E2F6 ChIP-seq with publicly available ChIP-seq datasets in mouse ESCs, in particular ChIP-seq of the PRC1.6 subunits. This revealed that E2F6 ChIP-seq profiles cluster with datasets of other components of the PRC1.6 complex including MAX, MGA, L3MBTL2, RYBP, and PCGF6 (Supplementary Fig. 1c). Indeed, E2F6 peaks in the genome show frequent binding of the other PRC1.6 subunits (Fig. 1f, g and Supplementary Fig. 1d). Furthermore, a comparison of the peaks of E2F6, MGA, and L3MBTL2 revealed a high degree of overlap (Supplementary Fig. 1e), further indicating that E2F6 and PRC1.6 frequently bind to the same genomic loci.

To identify E2F6-associated proteins, we generated ESCs stably expressing an hemagglutinin (HA)-tagged version of E2F6 (Fig. 1h) and performed immunoprecipitation on nuclear extracts followed by mass spectrometry (MS). We found a strong association of E2F6 with MGA, L3MBTL2, PCGF6, DP1, RING1A/B, and MAX (Fig. 1i), confirming that E2F6 associates with members of the PRC1.6 complex. RYBP was also detected below the threshold of significance (Supplementary Data 3). Other significant hits were SKT and KI-67, as well as cytoskeletal proteins that might be nonspecific contaminants (Supplementary Data 3). The interaction between HA-tagged E2F6 and endogenous MGA, PCGF6, and L3MBTL2 was confirmed by western blotting (Fig. 1j).

In summary, these results show that E2F6 is frequently bound to CpG-rich promoters and colocalizes with the PRC1.6 complex genome wide in mouse ESCs.

### E2F6 represses germline genes in mouse ES cells and in vivo.

To test whether E2F6 binding is causatively involved in transcriptional regulation, we performed RNA sequencing (RNA-seq) in $E2f6^{+/+}$ and $E2f6^{-/-}$ ESCs (Supplementary Data 1 and Supplementary Fig. 2a, b), and identified 187 genes significantly upregulated (fold change >3; adjusted $p$-value < 0.0001) upon *E2f6*-KO (Fig. 2a and Supplementary Data 4). Thus, most genes bound by E2F6 are not transcriptionally deregulated in $E2f6^{-/-}$ ESCs. In particular, the transcription of G1/S-induced E2F target genes known to be occupied by E2F6 (e.g., *Pcna*, *Rrm1*, *Cdc6*, and *Tk1*) was not affected in *E2f6*-KO cells (Supplementary Fig. 2c), consistent with previous findings in somatic cells[17]. Rather, GO analysis showed that the upregulated genes are enriched for GO terms related to germ cells (meiotic cell cycle $p = 7.62e - 12$; reproductive process $p = 2.73e - 9$, Supplementary Data 4). Twenty-four out of 37 genes that simultaneously have an E2F6 peak in their promoter and are upregulated in $E2f6^{-/-}$ ESCs are known germline-specific genes (Supplementary Fig. 2d),

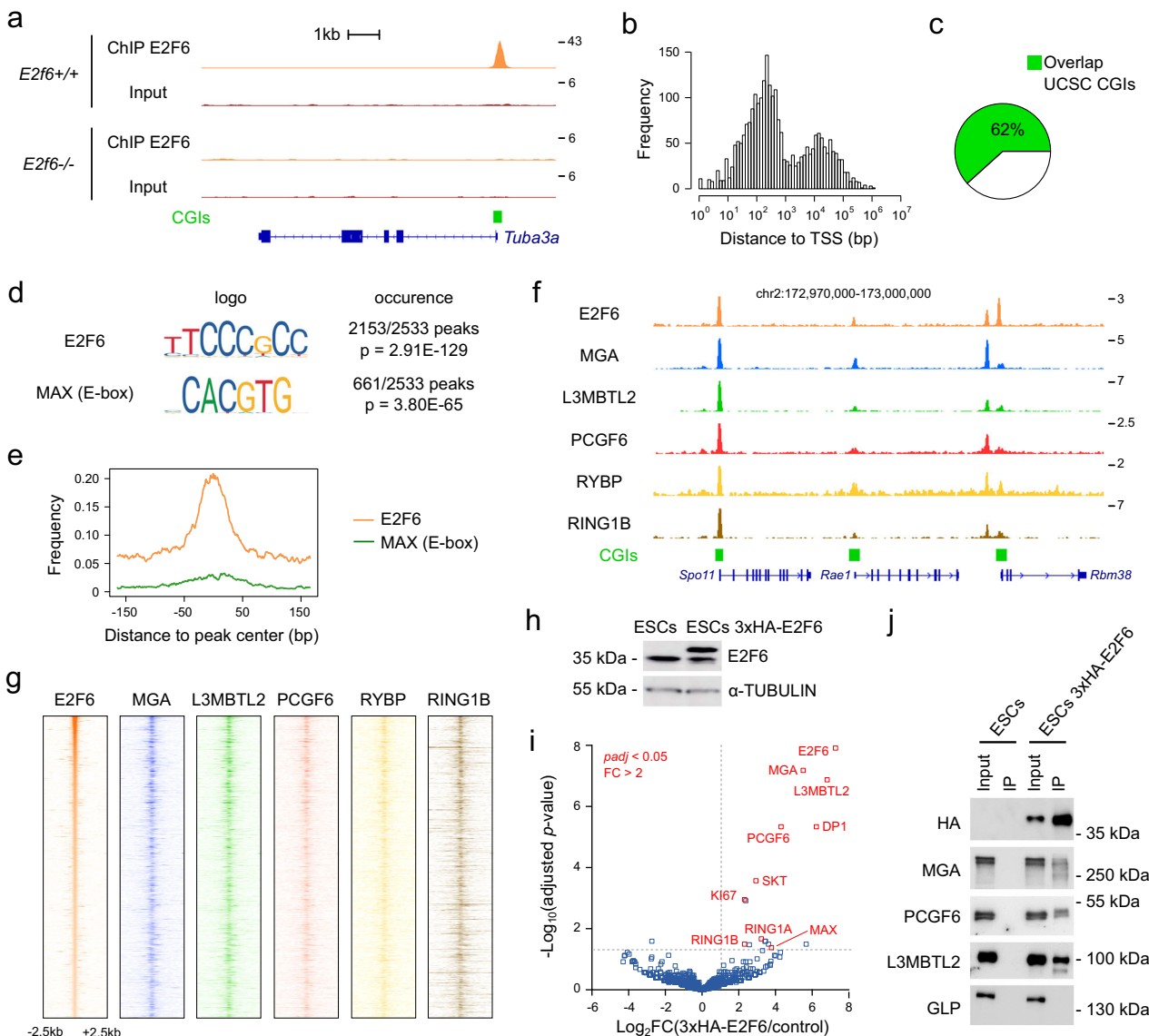

**Fig. 1 ChIP-seq analysis and interactome of E2F6 in ES cells. a** Genome browser tracks showing E2F6 ChIP-seq and input signals in *E2F6*[+/+] and *E2f6*[−/−] ESCs over the *Tuba3a* gene. UCSC CGI and RefSeq gene annotations are shown below the tracks. **b** Density histogram representing the distance of E2F6 peak summits to their closest TSS (*n* = 2533). **c** Percentage of E2F6 peaks overlapping with annotated UCSC CpG islands (CGIs). **d** The E2F6 and MAX E-box CACGTG motifs are the most enriched sequence motifs identified in E2F6 peaks (*p*-value: hypergeometric test). **e** Metaplot representing the position of E2F6 and MAX E-box motifs relative to the summits of E2F6 peaks (*n* = 2533). **f** Genome browser tracks of ChIP-seq signals in mouse ESCs showing that E2F6 peaks frequently colocalize with peaks of other PRC1.6 subunits. UCSC CGI and RefSeq gene annotations are shown below the tracks. **g** Heatmaps of E2F6, MGA, L3MBTL2, PCGF6, RYBP, and RING1B ChIP-seq read densities ±2.5 kb around E2F6 peak summits (*n* = 2533), ordered by E2F6 read densities. **h** Western blotting with an anti-E2F6 antibody showing the expression of the tagged E2F6 in ESCs transfected with 3×HA-E2F6. α-TUBULIN was used as a loading control. **i**. Volcano plot of E2F6 interaction partners in ESCs. Anti-HA immunoprecipitation was performed on nuclear extracts from control ESCs and ESCs expressing 3×HA-E2F6 (*n* = 5 independent experiments). Significantly enriched proteins are indicated in red. **j** Co-immunoprecipitation of E2F6 with PRC1.6 in mouse ESCs. Western blotting was performed on the input and anti-HA immunoprecipitate from control ESCs and ESCs expressing 3×HA-E2F6 using the indicated antibodies. GLP was used as a control nuclear protein that does not interact with E2F6. Source data are provided as a Source Data file.

indicating that germline genes are the main direct targets of E2F6. Genes upregulated in *E2f6*[−/−] cells without E2F6 binding probably reflect indirect or clonal effects. These germline genes include the previously known targets of E2F6 *Tuba3a*, *Slc25a31*, *Smc1b*, and *Stag3*[20,21,36] (Supplementary Fig. 2d, e). ChIP-seq profiles confirm that these germline genes have strong E2F6 signal in their promoter (Supplementary Fig. 3a). Furthermore, derepression of a subset of these germline genes in *E2f6*[−/−] ESCs was validated by reverse-transcriptase quantitative PCR (RT-qPCR) (Fig. 2b) and E2F6 binding was validated by ChIP-

qPCR at all germline gene promoters tested (Fig. 2c). This demonstrates that E2F6 plays a merely redundant role in the repression of most of its targets but is indispensable for silencing a group of germline genes in ESCs.

To test whether E2F6 plays similar functions in vivo, we analyzed embryos of *E2f6*-KO mice[37]. RNA-seq in *E2f6*[−/−] and control embryos dissected at E8.5 (Supplementary Data 1 and Supplementary Fig. 2a, b) identified 34 upregulated genes (fold change >3; adjusted *p*-value < 0.0001) (Fig. 2d and Supplementary Data 5). These upregulated genes were enriched for GO

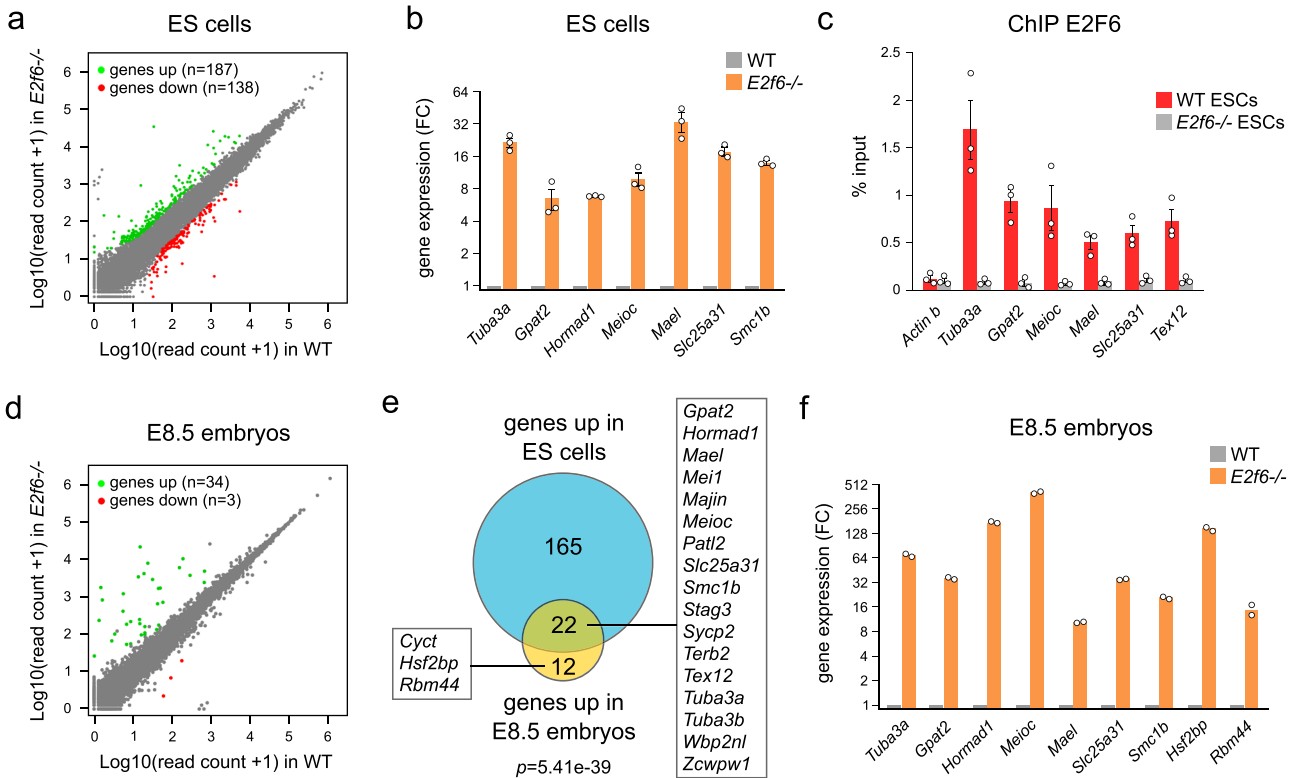

**Fig. 2 Identification of genes repressed by E2F6 in ES cells and embryos. a** Scatter plot comparing the normalized RNA-seq read counts for all RefSeq genes in $E2f6^{+/+}$ vs. $E2f6^{-/-}$ ESCs. Genes significantly upregulated or downregulated (fold change > 3; adjusted $p$-value < 0.0001) are highlighted in green and red, respectively. **b** Expression of E2F6 target genes by RT-qPCR in $E2f6^{-/-}$ ESCs. Shown is the fold change (FC) relative to WT cells (mean ± SEM, $n = 3$ independent experiments, expression normalized to *Gusb*, *Rpl13a*, and *Mrpl32*). **c** ChIP-qPCR analysis of the binding of E2F6 in the promoters of E2F6 target genes in WT and $E2f6^{-/-}$ ESCs (mean ± SEM, $n = 3$ independent experiments). **d** Scatter plot comparing the normalized RNA-seq read counts for all RefSeq genes in WT and $E2f6^{-/-}$ E8.5 embryos. Genes significantly upregulated or downregulated (fold change > 3; adjusted $p$-value < 0.0001) are highlighted in green and red, respectively. **e** Venn diagram comparing the genes upregulated in $E2f6^{-/-}$ ESCs and $E2f6^{-/-}$ embryos. The names of germline genes significantly upregulated in $E2f6^{-/-}$ ESCs and embryos or $E2f6^{-/-}$ embryos only are indicated. $p$-value: hypergeometric test. **f** Expression of E2F6 target genes by RT-qPCR in $E2f6^{-/-}$ embryos (mean fold change relative to WT, $n = 2$ embryos, expression normalized to *Actb* and *Rpl13a*). Source data are provided as a Source Data file.

terms related to germ cells (meiotic cell cycle $p = 6.96e - 16$; reproductive process $p = 3.82e - 9$, Supplementary Data 5) and significantly overlapped with the genes upregulated in ESCs (Fig. 2e, $p = 5.41e - 39$, hypergeometric test), revealing a core set of 17 germline genes repressed by E2F6 in ESCs and embryos, whereas a small number of germline genes was found strongly derepressed only in embryos (Fig. 2e). The derepression of germline genes in $E2f6^{-/-}$ embryos was validated by RT-qPCR (Fig. 2f). In summary, these results show that E2F6 is indispensable to repress a group of germline genes in ESCs and in vivo.

**E2F6 cooperates with MGA for the silencing of germline genes.** Germline genes repressed by E2F6 show binding of MGA, the other DNA-binding subunit of PRC1.6 previously implicated in the repression of germline genes in ESCs[30] (Supplementary Fig. 3a). As it is well known that ChIP peaks may result from indirect recruitment, we searched for E-box motifs recognized by MAX/MGA and found that several E2F6 target genes contain an E-box motif in their promoter (Fig. 3a). This raises the question whether E2F6 and MGA cooperate for the regulation of germline genes.

First, we compared the expression of germline genes in $E2f6^{-/-}$ and *Mga*-knockdown ESCs[30]. A comparative analysis revealed that most of the E2F6 target genes are concomitantly repressed by MGA (Fig. 3b), indicating that E2F6 and MGA share a set of

target genes related to germline functions in ESCs. MGA also represses additional germline genes that are not repressed by E2F6, such as *Tex101*, *Tdrd9*, *Taf7l*, *Mov10l1*, or *Tex19.2* (Fig. 3b and Supplementary Fig. 3b). However, as shown previously[33], we found that knockdown of MGA resulted in a strong reduction of E2F6 protein levels in ESCs (Supplementary Fig. 4a). This is due to protein destabilization at the posttranscriptional level, because MGA knockdown does not lead to reduced levels of *E2f6* mRNA transcripts (Supplementary Fig. 4b) and treatment with the MG132 proteasome inhibitor slightly increases the levels of E2F6 protein in MGA-kd ESCs (Supplementary Fig. 4a). To circumvent this confounding effect and test whether E2F6 and MGA synergize for the repression of germline genes, we depleted MGA by short hairpin RNA (shRNA) in $E2f6^{-/-}$ ESCs. Knockdown of MGA in $E2f6^{-/-}$ cells resulted in an additional derepression of the E-box-containing genes *Slc25a31* and *Tex12*, but did not further affect the expression of the E-box-lacking gene *Tuba3a* (Fig. 3c). In comparison to MGA, the knockdown of MAX had a weaker effect on the derepression of E-box-containing genes (Fig. 3c). Altogether, these results show that E2F6 and MGA cooperate for the silencing of germline genes with E-box-containing promoters.

Next, we investigated whether E2F6 participates in PRC1.6 recruitment to germline genes. Importantly, protein levels of PRC1.6 subunits are unaffected in *E2f6*-KO ESCs, suggesting that the PRC1.6 complex is intact in *E2f6*-null cells (Supplementary Fig. 4c). We performed ChIP-qPCR analysis of RYBP and PCGF6

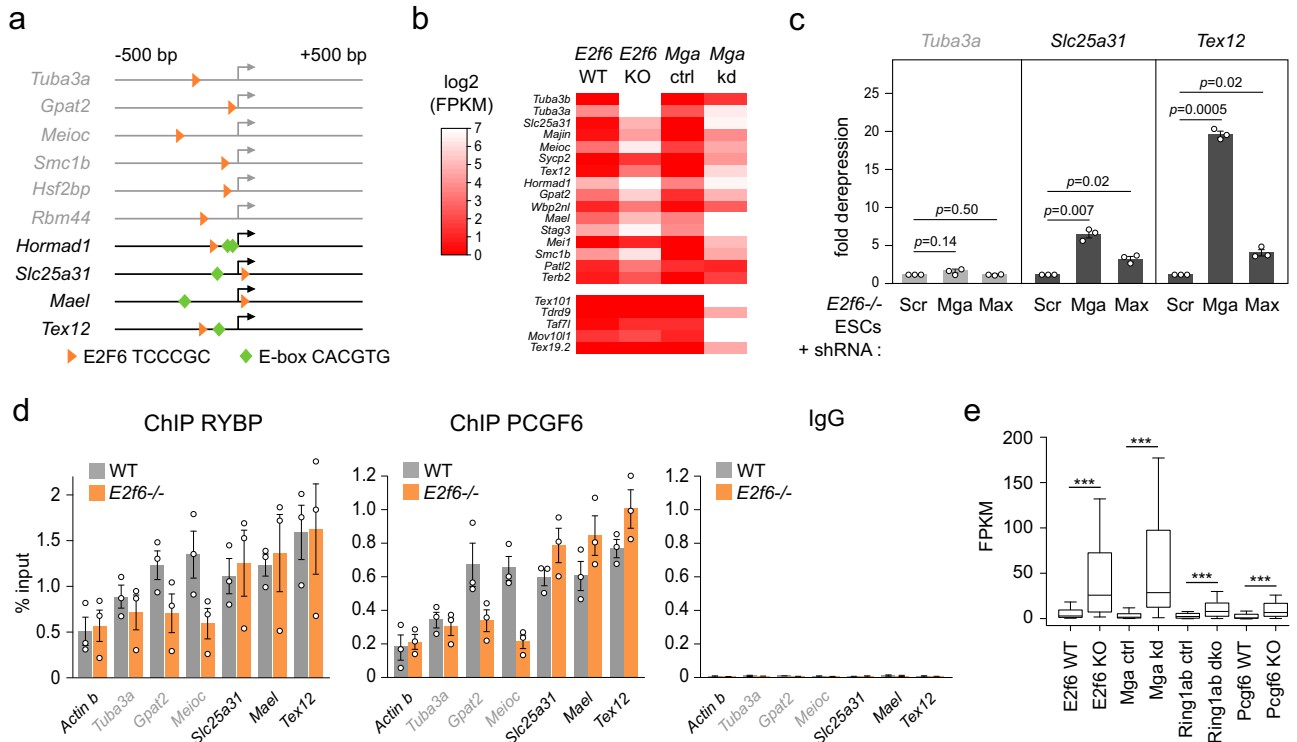

**Fig. 3 Cooperation between E2F6 and MGA in ES cells. a** Position of E2F6-binding sites (orange triangles) and MAX E-boxes (green diamonds) in the promoter regions (−500 to +500 bp relative to the start of transcription) of E2F6 target genes. The arrow indicates the position of the TSS. The genes containing only an E2F6 motif are written in gray. **b** Heatmap comparing the changes in gene expression of germline genes in *E2f6*−/− and *Mga*-knockdown ESCs[30]. MGA represses the E2F6 target genes (top of the heatmap) but also other germline genes not repressed by E2F6 (bottom of the heatmap). **c** RT-qPCR quantification of the expression of E2F6 target genes without (*Tuba3a*) or with E-boxes (*Slc25a31*, *Tex12*) in *E2f6*−/− ESCs expressing an shRNA against MGA, MAX, or a scramble control (shRNA Scr). The results are represented as a fold change relative to the scramble control (mean ± SEM, n = 3 independent experiments, expression normalized to *Actb*). p-values: two-tailed t-test. **d** ChIP-qPCR analysis of the binding of RYBP and PCGF6 to promoters of E2F6 target genes in WT and *E2f6*−/− ESCs. The results are represented as the percentage of signal relative to the input (mean ± SEM, n = 3 independent experiments). For each condition, the values from the ChIP (left) and IgG control (right) are shown. The genes containing only an E2F6 motif are written in gray. **e** Gene expression (plotted as FPKM values) of E2F6 target germline genes (n = 24 genes) in *E2f6*-KO, *Mga* knockdown, *Ring1a/b* DKO, and *Pcgf6*-KO ESCs[30] (paired Wilcoxon test, *E2f6*: p = 1.19e − 7, *Mga*: p = 2.38e − 7, *Ring1a/b*: p = 8.35e − 6, *Pcgf6*: p = 1.19e − 7). In the boxplots, the line indicates the median, the box limits indicate the upper and lower quartiles, and the whiskers extend to 1.5 IQR from the quartiles. Source data are provided as a Source Data file.

on six E2F6 target genes that either contain only an E2F6-binding motif in their promoter or both an E2F6 motif and E-box motif (Fig. 3a). Interestingly, RYBP and PCGF6 binding was reduced in *E2f6*−/− ESCs specifically in those promoters lacking E-box motifs (Fig. 3d, genes marked in gray). This suggests that E2F6 participates in the recruitment of PRC1.6, but that in the presence of E-boxes the MAX/MGA subunits are sufficient to tether PRC1.6 in the absence of E2F6 protein. Against our expectations, the undisturbed presence of PRC1.6 on E-box-containing gene promoters in *E2f6*−/− ESCs indicates that on these promoters, E2F6-dependent gene repression is decoupled from PRC1.6 recruitment. To further assess the contribution of PRC1.6 to the regulation of germline genes repressed by E2F6, we compared expression changes in *E2f6*-KO ESCs and ESCs inactivated for *Pcgf6* or *Ring1a/b*, the catalytic subunits of PRC1 complexes[30]. Interestingly, *Pcgf6*-KO and *Ring1a/b* double KO (DKO) ESCs undergo limited upregulation of E2F6 target genes compared to *E2f6*-KO and *Mga*-Kd ESCs (Fig. 3e). Collectively, these results suggest that E2F6 represses germline genes in part by PRC1.6-independent mechanisms.

**E2F6 function depends on its marked box domain.** We went on to investigate the role of E2F6 domains in the repression of germline genes by rescue experiments with wild-type (WT) and

mutant versions of E2F6 in *E2f6*−/− ESCs (Fig. 4a, b). Stable re-expression of WT HA-tagged E2F6 in *E2f6*−/− ESCs restored the repression of *Tuba3a* (Fig. 4c). In contrast, a mutant version of E2F6 (E68) carrying a point mutation of the DNA-binding domain[18] failed to bind to the *Tuba3a* promoter and rescue *Tuba3a* silencing (Fig. 4c). Previous studies showed that the marked box domain of E2F proteins mediate their target gene specificity[38]. To test the importance of the E2F6 marked box domain, we re-expressed a chimeric E2F6 harboring the marked box of E2F4 (E2F6-MB4, Fig. 4a), an E2F family member that can compensate the loss of E2F6 in cell cycle regulation[17]. This mutant E2F6 was able to bind to the *Tuba3a* promoter but failed to rescue *Tuba3a* repression (Fig. 4c). Thus, repression of the *Tuba3a* gene by E2F6 requires the DNA-binding and marked box domains of E2F6.

During cell cycle progression, activating E2Fs (such as E2F1) and E2F6 occupy E2F-responsive promoters sequentially[17,35]. Strikingly, we found that one distinctive feature of germline genes actively repressed by E2F6 is high E2F6 binding and reduced E2F1 binding (Fig. 4d), suggesting that E2F6 has the ability to specifically outcompete E2F1 binding at its target genes. To test this hypothesis, we analyzed E2F1 binding by ChIP-qPCR in the *Tuba3a* promoter. In *E2f6*−/− ESCs, the promoter of the derepressed *Tuba3a* gene becomes occupied by E2F1 (Fig. 4e).

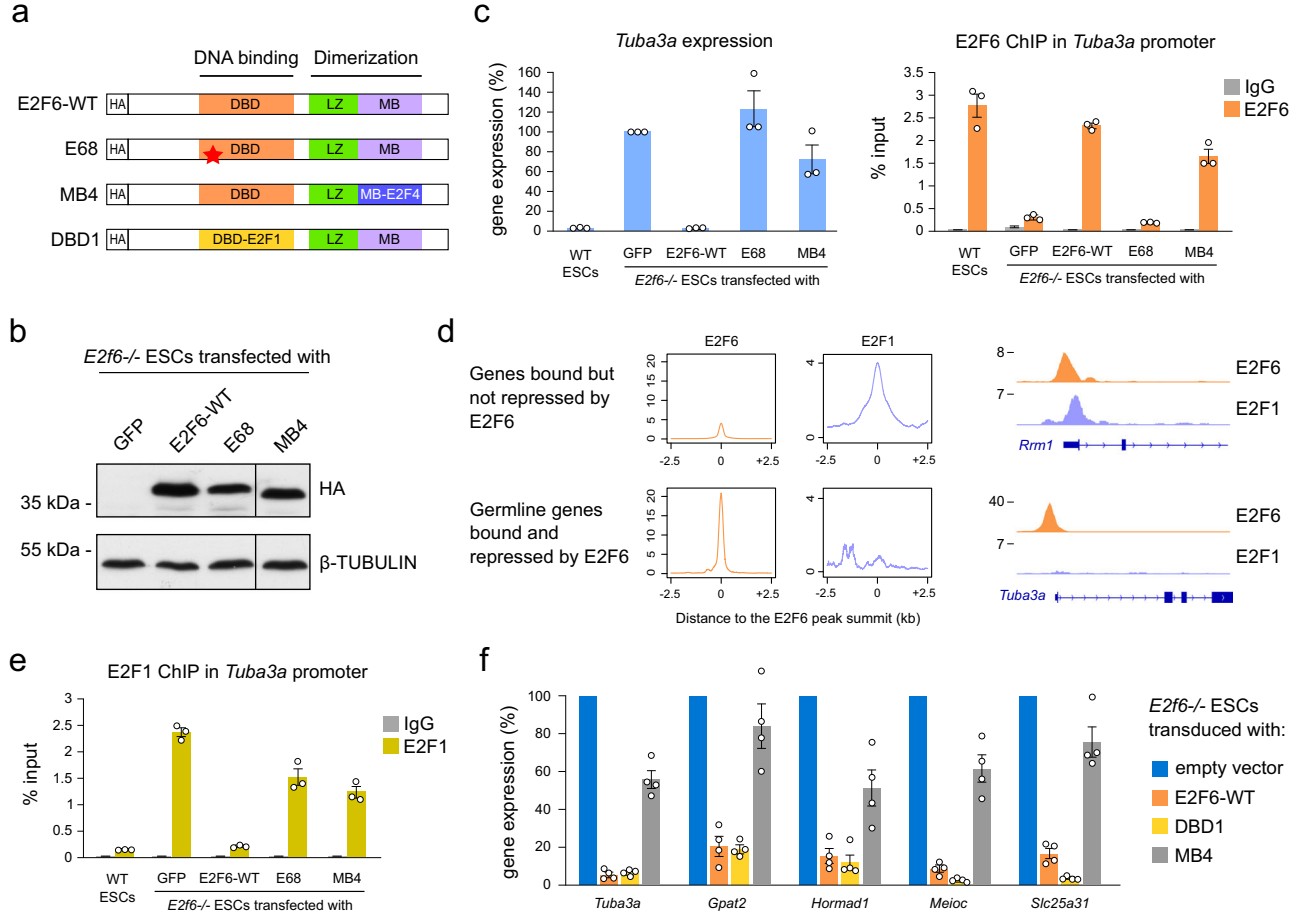

**Fig. 4 E2F6 function requires the DNA-binding and marked box domains. a** Schematic representation of the E2F6 protein and the mutants generated in the study. The E68 mutant contains a point mutation in the DNA-binding domain. DBD: DNA-binding domain; LZ: leucine zipper; MB: marked box; HA: HA tag. **b** E2F6 expression detected by western blotting with an anti-HA antibody in $E2f6^{-/-}$ ESCs rescued with an empty vector (GFP), E2F6 WT, E68 mutant or MB4 mutant. **c** RT-qPCR analysis of *Tuba3a* expression (left) and ChIP-qPCR analysis of E2F6 binding in the *Tuba3a* promoter (right) in WT ESCs and $E2f6^{-/-}$ ESCs rescued with an empty vector (GFP), E2F6-WT, E68 mutant or MB4 mutant. *Tuba3a* expression is represented as a percentage of expression relative to the GFP control (expression normalized to *Actb*). For ChIP experiments, the values for the E2F6 ChIP and IgG control are shown for each condition. All values are mean ± SEM ($n = 3$ experiments). **d** Metaplots showing E2F6 and E2F1 ChIP-seq read densities in E2F6 peaks linked to gene promoters bound but not repressed by E2F6 (top, $n = 1378$ genes) or promoters of germline genes bound and repressed by E2F6 (bottom, $n = 24$ genes). On the right are shown examples of E2F6 and E2F1 ChIP-seq profiles for the E2F cell cycle target gene *Rrm1* and the E2F6-repressed gene *Tuba3a*. **e** ChIP-qPCR analysis of E2F1 binding in the *Tuba3a* promoter in WT ESCs and $E2f6^{-/-}$ rescued ESCs (mean ± SEM, $n = 3$). **f** RT-qPCR analysis in $E2f6^{-/-}$ ESCs transduced with lentiviral vectors expressing E2F6-WT, E2F6-DBD1, or E2F6-MB4. The expression levels are shown as a percentage relative to cells transduced with an empty vector (mean ± SEM, $n = 4$, expression normalized to *Actb*). Source data are provided as a Source Data file.

Re-expression of exogenous WT E2F6 but not the E68 or MB4 mutants precluded E2F1 binding as efficiently as endogenous E2F6 in WT ESCs (Fig. 4e), demonstrating that E2F6 efficiently outcompetes E2F1 for promoter binding at the *Tuba3a* promoter. To investigate whether the ability of E2F6 to evict E2F1 results from different DNA-binding capabilities, we generated a swap-mutant of E2F6 (E2F6-DBD1) in which the DNA-binding domain of E2F6 was replaced by the corresponding E2F1 sequence (Fig. 4a). All attempts to establish stable transfected $E2f6^{-/-}$ ESC clones expressing E2F6-DBD1 failed and fluorescence-activated cell sorting analysis of transduced ESC pools revealed a strong counter selection against E2F6-DBD1-expressing cells (data not shown). However, expression analysis of cellular pools 4 days after lentiviral transduction into $E2f6^{-/-}$ ESCs demonstrates that the DNA-binding domains of E2F6 and E2F1 are functionally redundant in mediating repression of E2F6 target genes, in contrast to the E2F6-MB4 mutant that is unable to restore repression of germline genes (Fig. 4f). Therefore, the observed competitive advantage of E2F6 over E2F1 at germline genes is not linked to its DNA-binding domain and is mediated by the marked box domain.

Altogether, this shows that E2F6 target gene specificity and its ability to oppose activating TFs critically depend on its marked box domain.

**E2F6 represses germline genes by DNA methylation-independent mechanisms in early embryonic cells.** We then further explored the epigenetic mechanisms of E2F6-mediated repression of germline genes. Given that germline genes undergo long-term repression by promoter DNA methylation during development, we investigated the interplay between E2F6 and DNA methylation. First, we asked whether E2F6 represses germline genes in preimplantation embryonic cells prior to the phase of global DNA methylation establishment. To this end, we measured the expression of the E2F6 target genes *Tuba3a* and *Hormad1* in $E2f6^{+/+}$ and $E2f6^{-/-}$ blastocysts, and found that these genes are repressed by E2F6 in blastocysts (Fig. 5a), even though their promoters are hypomethylated at this stage[39]

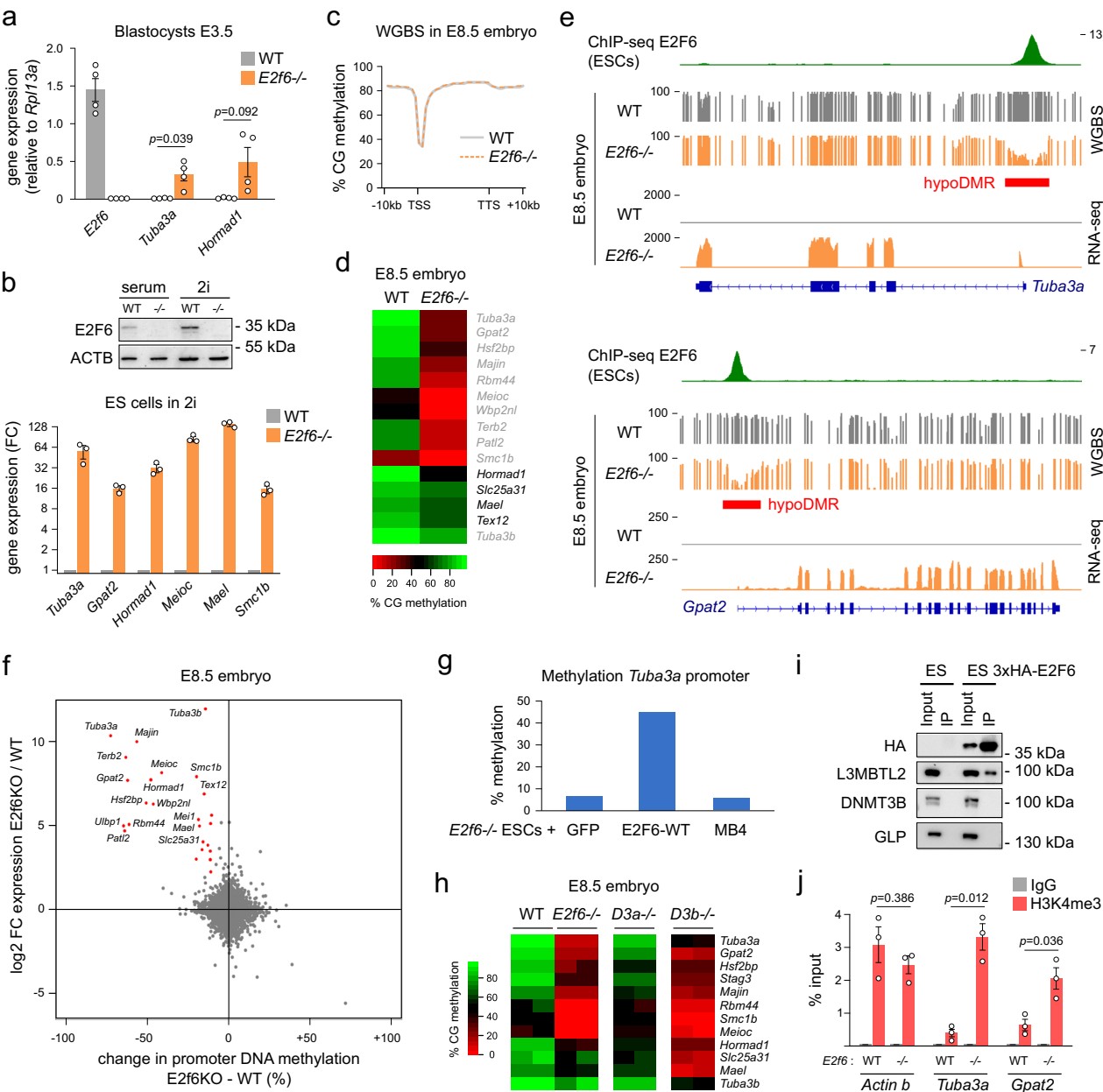

**Fig. 5 E2F6 facilitates de novo DNA methylation of germline genes during development. a** RT-qPCR analysis of the expression of *E2f6* and two E2F6 target genes in WT and *E2f6*$^{-/-}$ blastocysts (mean ± SEM, n = 4 blastocysts, expression normalized to *Rpl13a*). *p*-values: two-tailed *t*-test. **b** RT-qPCR analysis of E2F6 target genes in WT and *E2f6*$^{-/-}$ ESCs cultivated in 2i medium, represented as a fold change relative to WT (mean ± SEM, n = 3, expression normalized to *Gusb*, *Rpl13a*, and *Mrpl32*). A western blot showing E2F6 expression in ESCs cultivated in serum or 2i conditions is shown above the graph. **c** Metaplot showing the average percentage of CpG methylation measured by WGBS over RefSeq genes and 10 kb of flanking sequences in WT and *E2f6*$^{-/-}$ E8.5 embryos. **d** Heatmap representing the percentage of CpG methylation measured by WGBS in the promoters (−500 to +500 bp from the TSS) of E2F6 target genes in WT and *E2f6*$^{-/-}$ E8.5 embryos. The genes written in gray contain only an E2F6 motif in their promoter and the genes in black contain an E2F6 and E-box motif. **e** Genome tracks of E2F6 ChIP-seq (ESCs), WGBS, and RNA-seq in E8.5 embryos over two E2F6 target genes. Hypomethylated DMRs (hypoDMRs) are depicted by red rectangles. **f** Comparison of the changes in gene expression and promoter DNA methylation (measured by WGBS in −500 to +500 bp from the TSS) in *E2f6*$^{-/-}$ compared to WT E8.5 embryos for all genes. The genes simultaneously overexpressed (log$_2$FC > 2) and hypomethylated (< −10%) are highlighted in red. **g** Quantification of DNA methylation in the *Tuba3a* promoter by pyrosequencing in *E2f6*$^{-/-}$ ES cells rescued with an empty vector (GFP), E2F6-WT, or the E2F6-MB4 mutant. **h** Heatmap of RRBS methylation scores in the promoter of E2F6 target genes in E8.5 embryo knockout for *E2f6*, *Dnmt3a*, or *Dnmt3b*[3]. Each column represents an independent embryo. **i** Co-immunoprecipitation experiments to investigate the interaction between E2F6 and DNMT3B. Immunoprecipitation was performed with an anti-HA antibody on nuclear extracts from control ESCs and ESCs expressing 3×HA-E2F6. Western blotting was performed on the input and immunoprecipitates using anti-HA, anti-L3MBTL2, anti-DNMT3B, and anti-GLP antibodies. **j** ChIP-qPCR analysis of H3K4me3 levels in the promoters of *Actin b*, *Tuba3a*, and *Gpat2* in WT and *E2f6*$^{-/-}$ ES cells (mean ± SEM, n = 3). *p*-values: two-tailed *t*-test. Source data are provided as a Source Data file.

(Supplementary Fig. 5a). To corroborate this finding, we measured the expression of E2F6 target genes in $E2f6^{+/+}$ and $E2f6^{-/-}$ ESCs grown in 2i conditions, which inhibits DNA methylation[40]. Bisulfite sequencing confirmed that the promoters of E2F6 target genes are hypomethylated in 2i ESCs (Supplementary Fig. 5b) and E2F6 binding to germline genes is unchanged in 2i conditions as shown by a recent E2F6 ChIP-seq in 2i ESCs[41] (Supplementary Fig. 5c, d). Despite strongly reduced DNA methylation, germline genes remain repressed by E2F6 in 2i ESCs (Fig. 5b). Furthermore, RNA interference-mediated inhibition of *E2f6* leads to increased expression of germline genes in *Dnmts*-TKO ES cells that lack DNA methylation (Supplementary Fig. 5e). Collectively, these results suggest that E2F6 initially represses germline genes by mechanisms other than DNA methylation in early embryonic cells.

Previous studies identified HDAC1/2 and G9a as interactants of PRC1.6[23,24,32,42]; however, we found no enrichment of HDAC1/2 or G9a/GLP in the E2F6 interactome in ESCs (Supplementary Data 3), consistent with other reports[25,26,28]. Furthermore, almost all E2F6-repressed germline genes are not derepressed in *G9a* KO ESCs[43] (Supplementary Fig. 6a, b), and E2F6 and G9a repress distinct sets of genes in mouse embryos[44] (Supplementary Fig. 6c, d). In addition, most germline genes repressed by G9a were not identified as bound by E2F6, suggesting that they act in distinct pathways. Altogether, these data indicate that E2F6 function is independent of G9a.

To identify potential epigenetic mechanisms of E2F6-mediated repression, we examined which chromatin features distinguish germlines genes repressed by E2F6 from the other genes bound but not repressed by E2F6 in ES cells. This revealed that H3K9me3 and the H3K9 methyltransferase SETDB1[45–47] are specifically enriched in E2F6-repressed germline genes (Supplementary Fig. 6e), suggesting a link between E2F6-mediated repression and H3K9me3. To explore this possibility, we performed ChIP experiments and observed that the H3K9me3 mark is strongly reduced at E2F6 target genes in 2i $E2f6^{-/-}$ ESCs (Supplementary Fig. 6f). Furthermore, many E2F6-repressed germline genes are overexpressed upon SETDB1 inactivation in ESCs in 2i or serum conditions[48,49] (Supplementary Fig. 6a, b). This suggests that E2F6 represses germline genes in naive cells in part by favoring H3K9me3 deposition.

**E2F6 triggers DNA methylation of germline genes during development**. Next, we investigated whether E2F6 is required for the deposition of DNA methylation at germline genes after implantation by performing whole genome bisulfite sequencing (WGBS) in $E2f6^{+/+}$ and $E2f6^{-/-}$ E8.5 embryos (Supplementary Fig. 7a). Although global genome methylation is unchanged in $E2f6^{-/-}$ embryos (Fig. 5c), DNA methylation is specifically reduced in the promoter of several E2F6 target genes (Fig. 5d, e and Supplementary Fig. 7b). In total, WGBS identified 64 differentially methylated regions (DMRs) in $E2f6^{-/-}$ compared to $E2f6^{+/+}$ embryos, including 33 hypomethylated DMRs corresponding to many promoters of germline genes repressed by E2F6 (Supplementary Data 6). The hypomethylated DMRs colocalize with E2F6 ChIP-seq peaks and presumably reflect direct effects of E2F6 in recruiting DNA methylation, in contrast to hypermethylated DMRs that likely result from indirect effects (Supplementary Fig. 7c). The hypomethylation of germline genes was further validated by reduced representation bisulfite sequencing (RRBS) and combined bisulfite restriction analysis (COBRA) in independent embryos (Supplementary Fig. 7d–f). Correlation of gene expression and promoter DNA methylation changes showed that most genes derepressed in $E2f6^{-/-}$ embryos undergo concomitant promoter DNA hypomethylation (Fig. 5f),

indicating that E2F6 function is tightly linked to DNA methylation. Furthermore, most germline genes repressed by E2F6 are derepressed in *Dnmt3a/b* DKO embryos lacking de novo methylation[15] (Supplementary Fig. 7g, $p = 5.28e - 26$, hypergeometric test), demonstrating the importance of DNA methylation for E2F6-induced repression in embryos. In line with our previous results, germline genes containing only the E2F6 motif (e.g., *Tuba3a*, *Majin*, *Gpat2*, *Smc1b*, and *Meioc*) show an almost complete reduction of DNA methylation in $E2f6^{-/-}$ embryos, whereas genes containing an additional E-box (*Slc25a31*, *Hormad1*, *Mael*, and *Tex12*) show partial hypomethylation (Fig. 5d), suggesting that MAX/MGA compensate the absence of E2F6 for DNA methylation.

DNA hypomethylation of E2F6 target genes is also observed in $E2f6^{-/-}$ ESCs cultivated in serum plus leukemia inhibitory factor (LIF) (Supplementary Fig. 7e). Using this experimental system, we analyzed DNA methylation of the *Tuba3a* gene in $E2f6^{-/-}$ ESCs rescued with WT E2F6 or the MB4 mutant. WT E2F6 efficiently restored *Tuba3a* DNA methylation but not the MB4 mutant (Fig. 5g). This demonstrates that E2F6-induced DNA methylation requires the E2F6 marked box domain.

To further investigate whether DNA hypomethylation persists at later developmental stages, RRBS was performed in the brain, muscle, and kidney of $E2f6^{+/+}$ and $E2f6^{-/-}$ mice at 4 weeks of age (Supplementary Data 1). Promoter DNA hypomethylation of E2F6 target genes was detected in all organs at levels similar to the ones seen in E8.5 embryos (Supplementary Fig. 7h), demonstrating that the lack of de novo methylation in $E2f6^{-/-}$ embryos is not compensated throughout development.

To determine which DNMT mediates E2F6-dependent DNA methylation, we compared RRBS methylation levels of E2F6 target genes in $E2f6^{-/-}$ embryos with embryos knocked out for *Dnmt3a* or *Dnmt3b*[3]. E2F6 target genes are strongly hypomethylated in $Dnmt3b^{-/-}$ but not $Dnmt3a^{-/-}$ embryos (Fig. 5h), suggesting that DNMT3B is the main enzyme depositing DNA methylation at E2F6 target genes. Interestingly, DNMT3B is only slightly enriched in the E2F6 interactome below the threshold of significance (Supplementary Fig. 8) and we could not detect an E2F6–DNMT3B interaction by western blotting analysis of co-immunoprecipitation in ES cells (Fig. 5i). MS also did not reveal an interaction between E2F6 and DNMT3A, DNMT1, or UHRF1 (Supplementary Fig. 8). This suggests that E2F6 does not stimulate DNA methylation by direct interaction with the DNA methylation machinery. We then investigated whether E2F6 could indirectly influence DNA methylation by regulating H3K4 methylation. We found that germline genes repressed by E2F6 have low levels of promoter H3K4me3 compared to other genes bound but not repressed by E2F6 in ES cells (Supplementary Fig. 6e). Furthermore, E2F6 target genes *Tuba3a* and *Gpat2* show elevated levels of promoter H3K4me3 in $E2f6^{-/-}$ ES cells (Fig. 5j), suggesting that E2F6 could stimulate DNA methylation indirectly by maintaining low levels of H3K4 methylation.

Altogether, these results demonstrate that E2F6 is indispensable for promoting DNA methylation-dependent silencing to its target genes during embryonic development, apparently through indirect recruitment of DNMT3B.

**E2F6 is dispensable for the maintenance of epigenetic silencing in differentiated cells**. The general view is that CGI DNA methylation is not an initiating event in gene silencing but acts to lock in the silent state for long-term silencing. We thus speculated that E2F6 is required to initiate epigenetic silencing in early development but not to maintain it in differentiated cells. To test this hypothesis, we performed ectopic inactivation of *E2f6* by CRISPR-Cas9 in mouse embryonic fibroblasts (MEFs) and

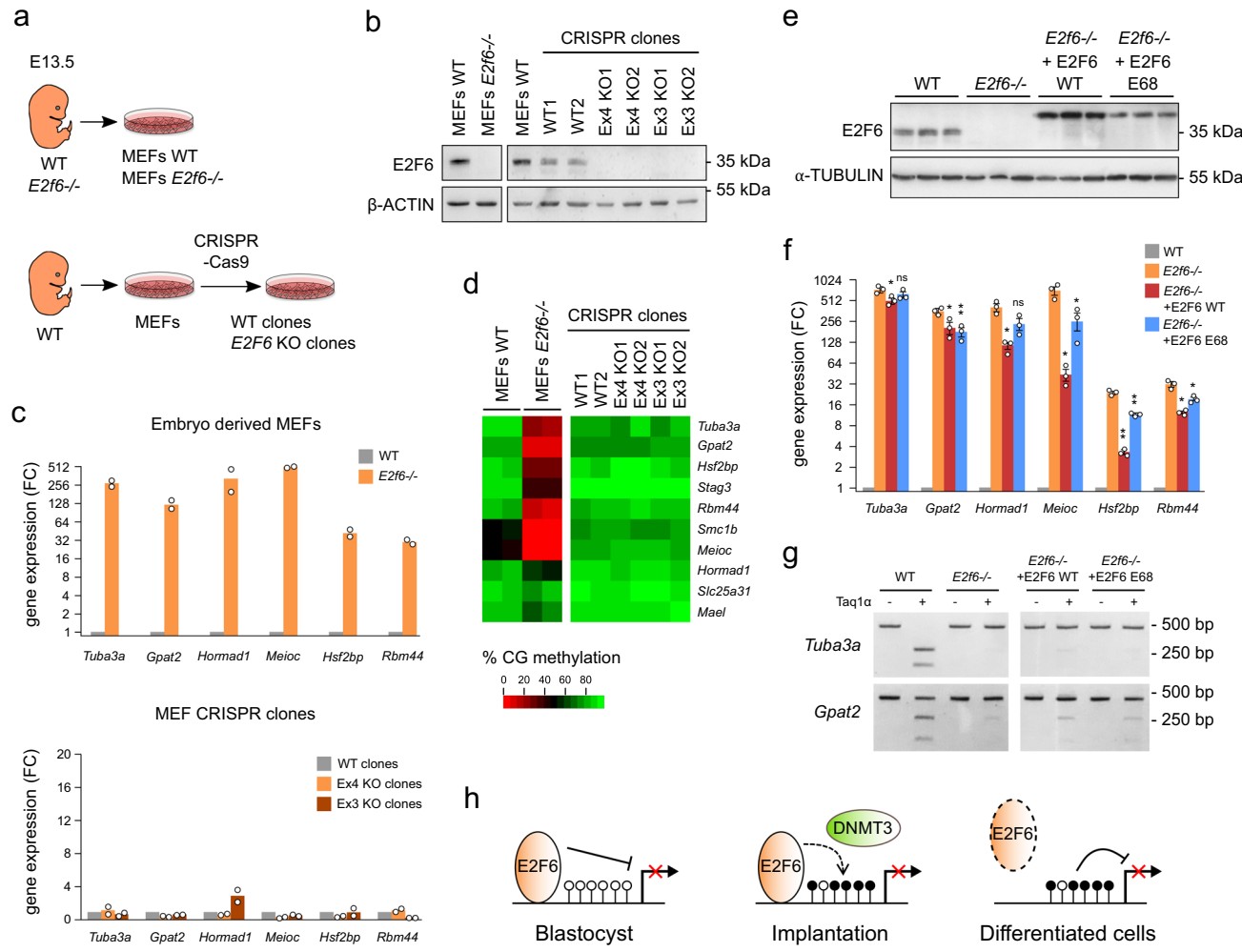

**Fig. 6 E2F6 is dispensable for the maintenance of epigenetic silencing. a** Experimental design. We either derived MEFs from WT and *E2f6*−/− embryos or generated E2f6-KO clones and WT control clones by CRISPR-Cas9. **b** Western blotting of E2F6 in MEFs derived from WT and *E2f6*−/− embryos and MEFs CRISPR-Cas9 clones. β-ACTIN was used as a loading control. **c** RT-qPCR analysis of the expression of E2F6 target genes in embryo-derived WT and *E2f6*−/− MEFs (top), and MEF CRIPSR-Cas9 clones (bottom) (mean fold change compared to WT controls, *n* = 2 independent cell lines or clones per condition, expression normalized to *Gusb*, *Rpl13a*, and *Mrpl32*). **d** Heatmap representing the percentage of CpG methylation profiled by RRBS in the promoters of E2F6 target genes in embryo-derived WT and *E2f6*−/− MEFs (left), and MEF CRISPR-Cas9 clones (right). **e** Western blotting of E2F6 in WT and *E2f6*−/− MEFs, and *E2f6*−/− MEFs transduced with lentiviruses expressing E2F6 or E2F6 E68 fused to 3×-HA. α-TUBULIN was used as a loading control. **f** RT-qPCR analysis of the expression of E2F6 target genes in WT and *E2f6*−/− MEFs, and *E2f6*−/− MEFs rescued with E2F6. The values are represented as the fold change compared to WT (mean ± SEM, *n* = 3 independent experiments, expression normalized to *Gusb*, *Rpl13a*, and *Mrpl32*). *$p < 0.05$, **$p < 0.01$, ns: not significant (two-tailed *t*-test compared to *E2f6*−/− cells). **g** Methylation analysis by COBRA of the *Tuba3a* and *Gpat2* promoters in WT and *E2f6*−/− MEFs, and *E2f6*−/− MEFs rescued with E2F6. **h** Model: E2F6 binds to and represses promoters of germline genes in preimplantation stages. At implantation, E2F6 recruits DNA methylation and initiates long-term epigenetic silencing of its target germline genes and subsequently becomes dispensable for the maintenance of the epigenetic repression in differentiated cells. Source data are provided as a Source Data file.

compared these cells to constitutive *E2f6*-KO MEFs derived from *E2f6*−/− embryos (Fig. 6a). It was previously shown by ChIP that E2F6 binds to the promoters of germline genes in MEFs[22]. CRISPR-Cas9-mediated KO of *E2f6* was performed by removing the exon 3 or exon 4, which creates a frameshift resulting in premature stop codons (Supplementary Fig. 9a). The inactivation of *E2f6* was validated by western blotting and RT-qPCR in two independent *E2f6*-KO clones for each condition (Fig. 6b and Supplementary Fig. 9b). *E2f6*-deficient MEFs generated by CRISPR-Cas9 did not reactivate E2F6 target genes, in contrast to MEFs derived from *E2f6*−/− embryos that showed a strong derepression of germline genes (Fig. 6c). Furthermore, RRBS showed that DNA methylation of E2F6 target genes is unaffected in CRISPR-Cas9 *E2f6*-deficient MEFs (Fig. 6d). In contrast to MEFs, ectopic inactivation of *E2f6* by CRISPR-Cas9 in ESCs leads

to derepression of germline genes (Supplementary Fig. 9c–e). To corroborate this finding in vivo, we monitored E2F6 expression after birth and observed that the E2F6 protein becomes barely detectable in various organs at 4 weeks of age compared to ES cells (Supplementary Fig. 9f). Despite no detectable E2F6, the silencing of germline genes is maintained in the liver and kidney (Supplementary Fig. 9g), further suggesting that the continuous presence of E2F6 is not required for the silencing of germline genes in organs. Altogether, these results demonstrate that E2F6 is necessary to initiate epigenetic silencing of germline genes in early embryonic cells but is no longer required for the maintenance of silencing in differentiated cells.

To examine whether the silencing of germline genes can be rescued by E2F6 in differentiated cells, MEFs derived from *E2f6*−/− embryos were stably transduced with a viral vector expressing

E2F6 or E2F6 mutated in the DNA-binding domain (Fig. 6e). Restoring E2F6 expression in E2f6−/− MEFs significantly reduced the expression of E2F6 target genes but was unable to restore efficient silencing at most tested targets (Fig. 6f). Furthermore, restoring E2F6 expression did not induce DNA methylation of the E2F6 target genes Tuba3a and Gpat2 (Fig. 6g). The inability of E2F6 to induce DNA methylation in MEFs might be due to the near absence of DNMT3B in these cells (Supplementary Fig. 9h).

Taken together, our results show that E2F6 is required to initiate stable epigenetic silencing of germline genes at a precise developmental window during early embryogenesis and subsequently becomes dispensable for maintaining the repressed state at later stages (Fig. 6h).

## Discussion

Genome-wide studies of DNA methylation during embryonic development revealed that CGI promoters are constitutively protected from DNA methylation, whereas de novo DNA methylation is specifically targeted to a small subset of promoter CGIs of germline genes[1–3]. However, the molecular pathways explaining this remarkable specificity are unknown. In this study, we extend previous work on the role of the transcription factor E2F6[20–22,36] and provide evidence that E2F6 is critical to promote DNA methylation and initiate lifelong epigenetic silencing of a subset of germline genes during mouse development.

Using genetic inactivation in mouse ES cells and in vivo, we demonstrate that E2F6 is critical to silence a subgroup of germline genes. E2F6 is a known DNA-binding component of the PRC1.6 complex; however, our results suggest that E2F6-dependent repression of germline genes is in part independent of PRC1.6 recruitment. Thus, E2F6 has both PRC1.6-dependent and -independent functions. In line with our results, a recent study showed that MAX induces epigenetic repression of germline genes in ESCs by PRC1-independent mechanisms[50].

An important question is why E2F6-mediated repression is restricted to a small subset of E2F6-bound promoters associated with germline genes. Germline genes contain canonical E2F6-binding motifs[20] but these are not specific for germline genes and occur in 2153 out of 2533 E2F6 ChIP-seq peaks. Therefore, the specificity of E2F6-mediated repression is more complex than simple DNA motif recognition. It has been shown previously for E2F and AP-1 TFs that their target specificity resides outside their DNA-binding domain[38,51]. In line with that, we demonstrated that the repressive function of E2F6 at germline genes depends on its marked box, a conserved domain that acts as a key determinant of target gene specificity in E2F TFs[38]. The marked box domain is not required for E2F recruitment to chromatin[52] but mediates protein–protein interactions[27]. We speculate that the E2F6 marked box domain could mediate local interaction with cofactors at the sites of germline promoters, which would escape detection in our interactome analysis.

Our developmental study showed that E2F6 has a dual function as follows: (1) it represses its target genes during the global hypomethylation phase in preimplantation stages and (2) it is essential to recruit DNA methylation during global genome de novo methylation at implantation, thus favoring a transition from DNA methylation-independent to DNA methylation-dependent repression of germline genes in development. The molecular basis for the E2F6 dependency of DNA methylation deserves further investigation. In a previous publication, an E2F6–DNMT3B interaction was described by overexpression in human HEK293 cells and in MEFs[22]. However, we were unable to detect a strong interaction between E2F6 and DNMT3B in mouse ES cells. Furthermore, to our knowledge, MS analysis of E2F6-containing PRC1.6 complexes so far failed to detect DNMT3A or

DNMT3B[23,24,26]. Overall, this does not support a direct recruitment of DNMT3 by interaction with E2F6. Instead, we favor the hypothesis that E2F6 indirectly facilitates DNA methylation deposition. E2F6-dependent H3K9me3 could recruit DNA methylation, which is consistent with SETDB1 being required for DNA methylation of germline genes in mouse ES cells[53]. Alternatively, increasing evidence supports the view that H3K4me3 and activating TFs play a key role in the protection of CGIs against DNA methylation[7–10,13]. Interestingly, we demonstrated that, whereas the majority of genomic E2F6 targets are bound by E2F6 and E2F1, E2F6 has the unique ability to preclude binding of E2F1 to germline genes. Therefore, E2F6 could override the intrinsic protection of CGIs against DNA methylation at germline genes by opposing transcriptional activators such as E2F1 and H3K4me3 deposition. E2F1 was previously linked to the protection against DNA methylation in oogenesis[54]. Future work should be aimed at testing these hypotheses.

Some E2F6 target genes are only partially hypomethylated in E2f6−/− embryos, suggesting that other pathways cooperate with E2F6 for DNA methylation recruitment at germline genes. As these genes contain promoter-proximal E-boxes, it is likely that E2F6 and MAX/MGA cooperate to induce epigenetic silencing of germline genes. In support of this model, DNA methylation of several germline genes is compromised in MAX-knockdown ESCs[50]. Unfortunately, the inability to maintain Mga−/− ESCs and the early lethality of Mga and Max KO embryos at pre- and peri-implantation stages[55,56] precludes a detailed study of the role of MAX/MGA in the epigenetic regulation of germline genes.

Once initiated, the epigenetic silencing of germline genes is stable and does not require the continued presence of E2F6. DNA methylation is a well-defined epigenetic mark mediating long-lasting gene repression and is one prime mediator of E2F6-induced stable silencing. Indeed, most of the E2F6 target genes are upregulated in DNA methylation-deficient embryos. However, our data suggest that E2F6 also induces long-term silencing of germline genes by other mechanisms than DNA methylation. Some target genes such as Smc1b and Meioc acquire only partial promoter CGI methylation and are much more derepressed in E2f6−/− embryos compared to Dnmt3a/b mutant embryos, suggesting that E2F6 induces long-term repression of germline genes through multiple epigenetic functions. These alternative mechanisms could involve polycomb-mediated H3K27 methylation, which is involved in silencing of the Smc1b gene in mouse ESCs[57].

Our study contributes to explain how epigenetic repression is targeted with remarkable specificity to germline genes during embryonic development, providing a paradigm for how transient repressive signals by DNA-binding factors are translated into lifelong repressive epigenetic signals. This is reminiscent of KRAB-ZFPs TFs establishing stable epigenetic repression of TEs in development[58], suggesting potential evolutionary convergences between the mechanisms that restrict expression of potentially harmful transposons and germline genes in mammalian somatic cells.

## Methods

**Mouse lines.** The E2f6-KO mouse line[37] was a gift from Jacqueline Lees (Cambridge, MA). Heterozygous mice were maintained in a C57BL/6J background and E2f6−/− animals were generated by mating heterozygous males and females. We used WT embryos from the same litters as controls. The morning of the vaginal plug was designed as E0.5. and embryos were dissected at E8.5 in M2 medium. Postnatal organs were collected at 4 weeks of age. Animal experimental procedures complied with the ethical regulations of the Comité d'Ethique Régional en Expérimentation Animale de Strasbourg (CREMEAS). Mice were housed with free access to food and water, a 12 h light/dark cycle, and controlled temperature (20–24 °C) and humidity (40–70%).

**ES cells.** *E2f6*[−/−] and *E2f6*[+/+] ESCs were established from blastocysts of *E2f6*[−/−] and control mice[21]. These cells were cultured on gelatin-coated plates in KO Dulbecco's modified Eagle's medium (DMEM) supplemented with 10% KO serum replacement, 5% fetal calf serum (Pan Biotech), 1000 U/mL LIF, 1× non-essential amino acids, 2 mM GlutaMAX, 1% penicillin/streptomycin, and 0.1 mM 2-mercaptoethanol. ESCs were adapted in 2i conditions in DMEM F12 medium supplemented with 50% Neurobasal medium, 1× N2 (Thermo Fisher Scientific), 1× B27 (Thermo Fisher Scientific), 2 mM L-Glutamine, 1% penicillin/streptomycin, 0.1 mM 2-mercaptoethanol, 0.5% bovine serum albumin (Merck), 1000 U/mL LIF, 1 μM PD0325901 (Merck), 3 μM Chir99021 (Merck), and 100 μg/mL Ascorbic Acid (Sigma). ES cells were cultured in 2i medium for 14 days. ES cells E14TG2a were cultured on gelatin-coated plates in DMEM GlutaMAX medium supplemented with 15% fetal bovine serum (FBS), 1000 U/mL LIF, 0.1 mM non-essential amino acids, 2 mM L-Glutamine, 1% penicillin/streptomycin, and 0.1 mM 2-mercaptoethanol. J1 WT and J1 Dnmt-TKO ES cells were a gift from M. Okano[59]. These cells were cultured on gelatin-coated dishes in DMEM Glutamax supplemented with 15% FBS, 1000 U/mL LIF, 0.1 mM non-essential amino acids, 1 mM sodium pyruvate, 50 U/mL penicillin/streptomycin, and 0.1 mM 2-mercaptoethanol. ESCs were passaged with Trypsin-EDTA 0.25% (Thermo Fisher Scientific) every 2 days. All ESCs were tested negative for mycoplasma contamination.

**E2f6 knockdown by siRNA.** J1 WT and J1 Dnmt-TKO ES cells were transfected with ON-TARGETplus SMART pool small interfering RNAs (siRNAs) (Horizon Discovery, non-targeting pool # D-001810-10 and E2f6 # L-047927) using Lipofectamine 2000 (Thermo Fisher Scientific). Lipofectamine 2000 (10 μL) and 0.5 mL DMEM were mixed with 200 pmol siRNA and 0.5 mL DMEM, and kept at room temperature (RT) for 20 min. The mix was added to a suspension of 1 mL of mESC (160 K cells/mL) in serum medium and plated on a 60 mm diameter petri dish. The medium was changed 6 h later and renewed every 24 h. Cells were collected 72 h after transfection for nucleic acid extraction. Target sequences for E2f6: 5′-GUUUAUGGAUCUCGUCAGA-3′/5′-CACAUUAGGUGGAUAGGAU-3′/5′-GUAUGUAACCUAUCAGGAU-3′/5′-UAAACAAAGUUGCCACAAA-3′.

**Isolation and culture of MEFs.** MEFs were isolated from E13.5 *E2f6*[−/−] and WT embryos. Primary fibroblasts were immortalized by infecting cells with retroviruses encoding the Antigen T produced in HEK293 cells with the plasmid pBABE-neo largeTcDNA (Addgene #1780). MEFs were grown in DMEM supplemented with 10% FBS and 1% penicillin/streptomycin in a humidified atmosphere containing 5% $CO_2$ at 37 °C. MEFs were passaged every 3 days and were tested negative for mycoplasma contamination.

**Inactivation of E2f6 by CRISPR-Cas9.** E2f6-KO cells were generated by deleting either the exon 3 or the exon 4 using two guide RNAs (gRNAs) flanking the exon. The gRNAs around exon 3 (gRNA1: 5′-CCCTCATTACTTCAGGATTG-3′; gRNA2: 5′-TAGTGCTGCCACTGCGCTAA-3′) and exon 4 (gRNA1: 5′-CTGA GAAATGTCGGTTCACA-3′; gRNA2: 5′-CATGGTGTGGAAAACTTAAC-3′) were designed with http://crispr.mit.edu. The gRNAs were cloned in the CRISPR-Cas9 plasmids PX458 (pSpCas9(BB)-2A-GFP, Addgene #48138) and PX459 (pSpCas9(BB)-2A-Puro, Addgene #62988). Immortalized MEFs were transfected by the PX458 plasmid using the Neon electroporation system (Thermo Fisher Scientific). Cells expressing green fluorescent protein (GFP) were selected 24 h post transfection by flow cytometry and plated at a clonal density in 96-well plates. Mouse ESCs E14TG2a (obtained from ATCC, CRL-1821) were transfected by the PX459 plasmid using Lipofectamine transfection reagent (Invitrogen) and selected for 72 h by Puromycin (2 μg/mL). Single ES colonies were selected and plated in 96-well plates. The clones were genotyped by PCR on genomic DNA isolated with the AllPrep DNA/RNA Mini Kit (Qiagen) using primers in the exon 3 or exon 4. We selected WT and *E2f6*-KO clones that were further validated by Sanger sequencing, western blotting, and RT-qPCR using primers spanning the exons 3 and 4 (provided in the Supplementary Data 7). For western blotting, whole cell extracts were prepared in RIPA lysis buffer (Cliniciences).

**shRNA knockdown of MGA and MAX in ES cells.** Lentiviral shRNA knockdown constructs were generated by cloning synthetic oligos corresponding to MGA shRNA (TRCN0000082083), MAX shRNA (TRCN0000039867), and scramble control of the Genetic Perturbation Platform (Broad Institute) into the pLKO.1 puro shRNA expression vector (Addgene #8453). Knockdown was performed by infecting *E2f6*[+/+] or *E2f6*[−/−] ESCs with lentiviral particles produced in HEK293T cells in the presence of 8 μg/ml polybrene. Twenty-four hours after transduction, ESCs were selected with puromycin (2 μg/ml) and RNA was collected after 48 h of selection.

**Expression vectors for ES cells.** For stable transfection experiments in ESCs, the sequences coding for E2F6 WT, as well as E68 and MB4 mutants, were inserted into the pCAG-EGFP-IB vector[60]. To generate the E68 DNA-binding mutant, the amino acids in positions 68 and 69 were changed from Leu-Val to Glu-Ser[18]. To generate the MB4 mutant, the E2F6 marked box domain (aa 180–240) was replaced by the marked box domain of E2F4 (aa 138–198). These constructs were

transfected into ESCs by electroporation with the Amaxa Nucleofector II device using the Amaxa Cell Line Nucleofector kit R and stable transfectants were selected with blasticidin (7.5 μg/ml). For the DNA-binding domain swap experiment, the sequences coding for E2F6 WT, DBD1, and MB4 mutants were inserted in the lentiviral expression vector pCDH-MSCV-MCS-EF1α-GFP + Puro (System Biosciences). The DBD1 mutant was generated by replacing the DNA-binding domain of E2F6 (aa 61–129) with the DNA-binding domain of E2F1 (aa 122–187). Viral particles were produced in HEK293T cells and ESCs were infected with 8 μg/ml polybrene. Twenty-four hours after transduction, puromycin (2 μg/ml) was added for 72 h before collecting the RNA samples.

**Expression vectors for MEFs.** The sequences coding for E2F6 WT and E68 fused to 3×-HA were cloned into the pMSCV Puro-IRES-GFP retroviral vector (Addgene #21654) with the In-Fusion HD cloning plus kit (Takara). Viral particles produced in HEK293T cells were used to infect immortalized *E2f6*[−/−] MEFs with 8 μg/ml polybrene. Twenty-four hours after infection, the cells were selected with Puromycin (2 μg/ml) for 48 h. The cells were collected 14 days after the infection for protein, DNA, and RNA extraction.

**Co-immunoprecipitation experiments.** ES cell line E14TG2a was transfected with the pCAG-3×HA-E2F6 vector and selected with blasticidine at 0.01 μg/μL until stable integration of the plasmid. Nuclear extract (1 mg) was incubated with anti-HA magnetic beads (Pierce) overnight at 4 °C. The complex bead–proteins was washed in a buffer containing 15 mM of Tris pH 8, 150 mM NaCl, 0.05% Triton. Proteins were eluted in Laemmli buffer (Thermo Fisher Scientific). As a control, immunoprecipitation was performed on E14TG2a cells not expressing 3×HA-E2F6. Twenty-five percent of eluted proteins and 15 μg of input proteins were loaded on a 10% SDS-polyacrylamide gel and assessed by western blotting.

**Interactome analysis by mass spectrometry.** Immunoprecipitations were carried out on 1 mg of nuclear extract from 3×HA-E2F6 and control E14TG2a cells with mMACS anti-HA microbeads (Miltenyi Biotec) according to the manufacturer's protocol. Protein complexes were eluted out of the magnetic stand with the SDS gel-loading buffer from the kit and eluted proteins were precipitated overnight with five volumes of cold 0.1 M ammonium acetate in 100% methanol before being digested with sequencing-grade trypsin (Promega). Generated peptides were analyzed by nanoscale liquid chromatography-MS/MS on a QExactive Plus mass spectrometer coupled to an EASY-nanoLC-1000 (Thermo Fisher Scientific) and they were identified with the Mascot algorithm (version 2.5, Matrix Science) using the Swissprot database with the *Mus musculus* taxonomy (release 2019_10) and the software's decoy strategy. Mascot identifications were imported into the Proline software[61] v1.4 where they were validated using the following settings: Mascot pretty rank equal to 1 and 1% false discovery rate on both peptide spectrum matches (PSM score) and protein sets (Protein Set score). The total number of MS/MS fragmentation spectra was used to quantify each protein. For the statistical analysis of the data, we compared the co-immunoprecipitation data collected for 3×HA-E2F6 samples against the negative controls using R v3.5.0. The spectral counts were normalized with DESeq2 (median of ratios method) and EdgeR was used to perform a negative-binomial test and calculate the fold change, *p*-value, and adjusted *p*-value corrected by Benjamini–Hochberg for each protein. We identified statistically enriched protein partners with a fold change >2 and an adjusted *p*-value < 0.05.

**ChIP of E2F6, E2F1, PRC1.6, and histone modifications.** ESCs were crosslinked in phosphate-buffered saline (PBS) with 1% formaldehyde for 15 min on ice and quenched with 125 mM glycine for 5 min. Cells were lysed in 50 mM Tris pH 8, 10 mM EDTA, 1% SDS with 1× cOmplete Mini protease inhibitor cocktail (Sigma Aldrich) and sonicated with a Bioruptor (Diagenode) at 4 °C. Eighty micrograms of chromatin were used per ChIP reaction and incubated with antibodies pre-bound to Dynabeads protein G (Thermo Fisher Scientific) overnight at 4 °C. The following antibodies were used: E2F6 (ab53061, Abcam), PCGF6 (24103-1-AP, Proteintech), RYBP (AB3637, Millipore), E2F1 (C-20, Santa Cruz Biotechnology), H3K4me3 (003-050, Diagenode), and as control normal mouse IgG (12-371, Millipore). The beads were washed successively in low-salt buffer (0.1% SDS, 1% Triton X-100, 2 mM EDTA, 20 mM Tris pH 8, 150 mM NaCl), high-salt buffer (0.1% SDS, 1% Triton X-100, 2 mM EDTA, 20 mM Tris pH 8, 500 mM NaCl), LiCl-buffer (0.25 M LiCl, 1% NP40, 1% sodium deoxycholate, 1 mM EDTA, 10 mM Tris pH 8), and TE (10 mM Tris pH 8, 1 mM EDTA). Bound chromatin was eluted with elution buffer (1% SDS, 0.1 M NaHCO₃) and crosslinking was reverted at 62 °C overnight in the presence of proteinase K (Qiagen). DNA was purified with a PCR Purification Kit (Qiagen). For ChIP of H3K9me3, ESCs were crosslinked in PBS with 1% formaldehyde for 10 min at RT and quenched with 125 mM glycine for 5 min. Cells were lysed in a buffer containing 0.1% SDS, 1% Triton X-100, 1 mM EDTA, 50 mM HEPES-KOH, 140 mM NaCl, 0.1% sodium deoxycholate, 1× protease inhibitor cocktail (Santa Cruz). The chromatin was sonicated with a Bioruptor (Diagenode) at 4 °C. Antibodies were pre-bound to protein A-coated magnetic beads (Diagenode) and incubated overnight at 4 °C with chromatin from 200,000 cells per ChIP reaction. We used antibodies against H3K9me3 (ab8898, Abcam), H3 (07-690, Merck), and IgG2A as a control (MAB0031, R&D Systems). Chromatin was

successively washed in three buffers containing 0.1% SDS, 1% Triton X-100, 2 mM EDTA, 20 mM Tris pH 8, 150 mM NaCl, then 0.1% SDS, 1% Triton X-100, 2 mM EDTA, 20 mM Tris pH 8, 500 mM NaCl and 0.25 M LiCl, 1% NP40, 1% sodium deoxycholate, 10 mM Tris pH 8, 1 mM EDTA. Chromatin was decrosslinked for 4 h at 65 °C and then treated with proteinase K (Euromedex) for 1 h at 56 °C. DNA was isolated by phenol–chloroform extraction and ethanol precipitation for qPCR analysis. The primers are provided in the Supplementary Data 7.

**ChIP-seq analysis**. ChIP-seq libraries of E2F6 ChIP and input samples were prepared from $E2f6^{+/+}$ and $E2f6^{-/-}$ ESCs using the DNA sample kit (Illumina) following the manufacturer's instructions. After adapter ligation, the DNA was PCR amplified with Illumina primers for 15 cycles and library fragments of ~250 bp were band isolated from an agarose gel. Libraries were sequenced in single-end $1 \times 50$ bp with an Illumina HiSeq 1500. For all ChIP-seq datasets used in the study, reads were trimmed using Trim Galore v0.4.4, aligned to the mouse genome assembly mm10 using bowtie2 v2.3.0, and only reads with a mapping quality above 10 were kept. Read density tracks were generated using genomeCoverageBed from bedtools, from reads extended to 200 bp and visualized using the IGV browser. Peaks were called using Peakzilla with default settings. The analysis of motif enrichment and distribution of motifs in peaks was performed using TFmotifView[62]. Briefly, control regions were selected randomly within the same chromosome from regions with matched CpG content within DNase open-chromatin regions. E2F6 peak regions (150 bp around peak summits) and shuffled control regions were searched for motif occurrences using MAST v5.1.0 (from the MEME suite) with a dynamic $p$-value threshold based on the motif information content (IC) ($p$-value $= 1/2IC$). The statistical significance of the motif enrichment in peaks over control regions was assessed using a hypergeometric $p$-value. The overlap of E2F6 peaks with CGIs was done by intersecting peak summits with the CGI annotation downloaded from the UCSC mm10 reference genome. For each E2F6 peak summit, the distance to their closest TSS was calculated using the UCSC Refseq annotation. To identify E2F6-bound gene promoters, we selected the genes corresponding to E2F6 peak summits distant <500 bp from their closest Refseq TSS. GO analysis of E2F6-bound promoters was performed using DAVID v6.8. Average ChIP-seq signals around E2F6 peak summits were calculated using bwtool v1.0. The pairwise correlation of ChIP-seq read densities was calculated by merging both peak regions with mergedBed from bedtools, extracting the maximum read density for both samples using bwtool summary, and their correlation was represented using the aheatmap function from the NMF package v0.21.0 in R. The overlap of E2F6, MGA, and L3MBTL2 peaks was generated with the findOverlapsOfPeaks function from the ChIPpeakAnno package v3.18.2 in R with a maximum gap of 500 bp. The datasets used for ChIP-seq analysis are described in the "Data availability" statement. For these datasets, raw reads were processed using the same pipeline as for E2F6.

**Transcriptome analysis by RNA-seq**. RNA-seq was performed on three independent cultures of ES cell lines and three single WT and $E2f6^{-/-}$ embryos collected from the same litters. RNAs from ESCs were extracted using the RNeasy kit (Qiagen). For embryos, we simultaneously prepared genomic DNA and total RNA from whole single embryos with the AllPrep DNA/RNA Mini Kit (Qiagen). RNA quality was verified on a 2100 Bioanalyzer (Agilent). RNA-seq libraries were generated from 250 ng of total RNA using the TruSeq Stranded Total RNA Library Prep kit with Ribo-Zero depletion (Illumina) according to the manufacturer's instructions, and sequenced in paired-end $2 \times 100$ bp on an Illumina HiSeq4000. Reads were mapped using TopHat v2.0.13 with a RefSeq transcriptome index and counted in RefSeq genes with HTSeq v0.7.2 (parameters –t exon –s reverse). Differentially expressed genes were identified using DESeq2 (fold change >3, adjusted $p$-value < 0.0001). Genes from the Y chromosome were excluded. GO analysis was performed with DAVID 6.8 (https://david.ncifcrf.gov). We also used published RNA-seq datasets, which are described in the "Data availability" statement.

**Methylome analysis by RRBS and WGBS**. RRBS libraries were prepared by $Msp$I digestion from 50 to 100 ng genomic DNA[44] and sequenced in paired-end $2 \times 75$ bp on an Illumina HiSeq4000 at Integragen SA (Evry, France). We trimmed RRBS reads to remove low-quality bases with Trim Galore v0.4.2 and aligned reads to the mm10 genome with BSMAP v2.74 (parameters -v 2 -w 100 -r 1 -x 400 -m 30 -D C-CGG -n 1). We calculated methylation scores using methratio.py in BSMAP v2.74 (parameters -z -u -g) and filtered CpGs covered by a minimum of eight reads. For WGBS, 100 ng genomic DNA was fragmented to 350 bp using a Covaris E220 sonicator and bisulfite-converted with the EZ DNA Methylation-Gold kit (Zymo Research). WGBS libraries were prepared using the Accel-NGS Methyl-Seq DNA Library Kit and the Methyl-Seq Set A Indexing Kit (Swift Biosciences) according to the manufacturer's instructions, with six PCR cycles for the final amplification. The libraries were purified using Ampure XP beads (Beckman Coulter) and sequenced in paired-end $2 \times 100$ bp with an Illumina HiSeq4000 at Integragen SA (Evry, France). The reads were trimmed to remove low-quality bases and the first five bases of reads R1 and ten bases of reads R2 using Trim Galore v0.4.2 (parameters -q 20 --clip_R1 5 --clip_R2 10), and aligned to the mouse mm10 genome using Bismark v0.18.2 with default parameters. Methylation calls were

extracted using Bismark v0.18.2 and only CpGs covered by more than five reads were retained for analyses. Metaplots of methylation over genes were generated by calculating the percentage of CpG methylation in 20 equal-sized windows within each RefSeq gene and ten 1 kb windows of flanking sequences. DMRs were identified using eDMR from the methylKit R package with a minimum of seven differentially methylated CpGs, a difference in methylation >20% and a $q$-value < 0.001.

**COBRA and bisulfite sequencing**. Genomic DNA (100 ng) was bisulfite-converted with the EpiTect bisulfite kit (Qiagen). The target regions were amplified by touchdown PCR with the Platinum Taq DNA Polymerase (Thermo Fisher Scientific) using the following conditions: 20 cycles of 30 s at 95 °C, 30 s at 60–50 °C (with a 0.5 °C decrease per cycle), 50 s at 72 °C followed by 35 cycles of 30 s at 95 °C, 30 s at 50 °C, and 50 s at 72 °C. The PCR products were purified using the PCR cleanup kit (Macherey Nagel). For COBRA, 40 ng of PCR product were digested by Taq1α (Thermo Fisher Scientific) and loaded on an agarose gel alongside 40 ng of undigested PCR product. For bisulfite sequencing, the PCR products were cloned by TA cloning in the pCR2.1 vector (Thermo Fisher Scientific) and sequenced by Sanger sequencing (Eurofins). Sequences were aligned with the BISMA software and filtered to remove identical clones. The primers are provided in the Supplementary Data 7.

**Pyrosequencing**. Genomic DNA was bisulfite-converted with the EpiTect bisulfite kit (Qiagen). The target region was amplified by PCR using the ZymoTaq PreMix (Epigenie) and pyrosequencing was performed by Varionostic (Ulm, Germany). The primers are provided in the Supplementary Data 7.

**Reverse-transcriptase quantitative PCR**. Total RNAs were reverse transcribed into cDNA with the Maxima first strand cDNA synthesis kit (Thermo Fisher Scientific). qPCR was performed with the KAPA SYBR FAST qPCR kit (KAPA Biosystems) on a StepOnePlus realtime PCR system (Applied Biosystems). The expression of target genes was normalized with several housekeeping genes (*B2m*, *Actin b*, *Gusb*, *Rpl13a*, or *Mrpl32*) as indicated in the figure legends. No-RT controls were tested to exclude the presence of contaminating DNA. The primer sequences for qPCR are provided in the Supplementary Data 7.

**Gene expression analysis in single blastocysts**. Blastocysts were isolated from uterine tubes in M2 medium and washed in PBS 1× droplets. Single blastocysts were collected in microtubes in 30 μL of $H_2O$ supplemented with RNase inhibitor (Promega) and lysed by performing five rounds of freezing in liquid nitrogen and thawing at 37 °C for 5 min. RNAs were reverse transcribed into cDNA with the Maxima first strand cDNA synthesis kit (Thermo Fisher Scientific) for qPCR analysis of target genes and *Rpl13a* as a normalization control. The blastocysts were genotyped by expression of the *E2f6* transcript. The primers are provided in the Supplementary Data 7.

**Western blotting**. Proteins were run on SDS-polyacrylamide gels and transferred to 0.2 μm nitrocellulose membranes using the Trans-blot Turbo Blotting System (Biorad). The membranes were blocked with Tris Buffered Saline (TBS), 5% milk, 0.05% Tween for 1 h at RT, incubated with the primary antibodies and then with horseradish peroxidase-conjugated secondary antibodies followed by chemiluminescence detection using the ECL detection reagent (Amersham, GE Healthcare). We used the following antibodies: E2F6 (Kerafast LLF6-2, 1 : 500), ACTB (Sigma Aldrich A2066, 1 : 1000), α-TUBULIN (Sigma Aldrich T9026, 1 : 1000), β-TUBULIN (Sigma Aldrich T8328, 1 : 1000), RYBP (Millipore AB3637, 1 : 1000), L3MBTL2 (Active Motif 39570, 1 : 500), PCGF6 (Proteintech 24103-1-AP, 1 : 500), MGA (Thermo Fisher Scientific PA5-59934, 1 : 500), RING1B (MBL D139-3, 1 : 1000), MAX (Santa Cruz Biotechnology sc-197, 1 : 200), GLP (R&D Systems PP-B0422-00, 1 : 500), DNMT3B (Abcam ab13604, 1 : 250), and HA (Roche 3F10, 1 : 500).

**Statistics and reproducibility**. The experiments were repeated independently as described in the figure legends. We performed five independent replicates for the interactome analysis and three independent replicates for RNA-seq. In vivo measurements were performed on independent animals. Western blottings were performed at least twice with similar results and one representative image is shown in the figures. Details on the statistical tests are given in the figure legends.

**Reporting summary**. Further information on research design is available in the Nature Research Reporting Summary linked to this article.

## Data availability
The RNA-seq, ChIP-seq, RRBS, and WGBS data generated in this study are available in the NCBI Gene Expression Omnibus (GEO) under the accession number GSE149025. The following published ChIP-seq datasets were used: KDM2B (GSM1272789), EZH2 (GSM1917297), SUZ12 (GSM1917296), RING1B (GSM1917298), G9a (GSM1215219), RYBP (GSM1041375), PCGF6 (ERX2161380), L3MBTL2 (ERX2161378), MGA

(ERX2161379), MAX (GSM1171650), ZFX (GSM288352), E2F1 (GSM288349), CTCF (GSM747534), TIP60 (GSM1183114), DMAP1 (GSM1183112), MYC (GSM1183111), TET1 (GSM611192), HDAC2 (GSM687279), ESRRB (GSM288355), DAX1 (GSM1183116), OCT4 (ERX1965633), KLF4 (ERX1965624), NANOG (GSM288345), SOX2 (GSM1910640), E2F6 in 2i conditions (GSM2907075), SETDB1 (GSM1893615, GSM440256), H3K9me3 (GSM1893613, GSM1487056), and H3K4me3 (GSM1487054). We also used RNA-seq datasets from *G9a* KO embryos (GSE71500), G9a KO ESCs (GSE49669), and SETDB1 KO ESCs (GSE29413, PRJNA544540). The mass spectrometry proteomics data have been deposited to the ProteomeXchange Consortium via the PRIDE partner repository with the dataset identifier PXD023974. We used the following publicly available databases: UCSC genome annotations, Swissprot database. All relevant data are available in the Supplementary Information files or from the corresponding authors. The Source data for Figs. 1h, j, 2b, c, f, 3c–e, 4b, c, e, f, 5a, b, h, i, and 6b, c, e–g, and Supplementary Figs. 2c, e, 4a–c, 5e, 6f, 7f, and 9c–h are provided with this paper in the source data file.

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

## Acknowledgements

We are grateful to Dr. Jacqueline Lees for providing *E2f6*-knockout mice. This work was supported by the European Research Council (ERC Consolidator grant number 615371 to M.W.), the Agence Nationale de la Recherche (ANR), the Institut National du Cancer (INCa), the Deutsche Forschungsgemeinschaft (DFG, grant number 63465200 to C.H.), and the ITI InnoVec of the University of Strasbourg, CNRS and Inserm (IdEx ANR-10-IDEX-0002, SFRI ANR-20-SFRI-0012). T.D. was recipient of a Doctoral fellowship from the French Ministry for Higher Education and Research. Mass spectrometry instrumentation was granted by the University of Strasbourg (IdEx Equipement mi-lourd 2015).

## Author contributions

T.D., M.T., U.F., H.A.A., J.V., J.C., P. Hammann, S.K., U.L., G. Alvarez, G. Auclair, and J.H. performed the experiments. A.F.B., M.D., P. Hansen, and P.N.R. performed bioinformatic analysis of sequencing data. J.C. performed the bioinformatic analysis of mass spectrometry data. M.T., C.H. and M.W. designed the study and supervised the experiments. M.T. and M.W. wrote the manuscript.

## Competing interests

The authors declare no competing interests.
