## [Peer Review File · Nature Communications]

REVIEWER COMMENTS

Reviewer #1 (Remarks to the Author):

The expression of germline genes involved in meiosis and gametogenesis is repressed by multiple mechanisms in early embryonic and somatic cells. For example, the non-canonical PRC1.6 complex, the histone methyltransferases G9A/GLP, as well as SETDB1, repress both unique and overlapping sets of germline genes in pluripotent mouse ESCs through the accumulation of H2AK119ub1 and H3K9me2/3, respectively at their promoter regions (Ehdoh et al. 2017, eLife; Maeda et al. 2013, Nat Commun; Karimi et al. 2011, Cell Stem Cell), guided in part by the DNA binding factors MGA/MAX. In mouse somatic cells, the transcription factor E2F6 was reported to have a related role in the silencing of such germline genes via the recruitment of the DNA methyltransferase DNMT3B, followed by the deposition of CpG methylation (Velasco et al. 2010, PNAS). However, these findings were predominantly derived from somatic cells of a late embryonic stage, or for other PRC1.6 subunits, ESCs. Therefore, the "order of events" and function(s) of each chromatin factor/complex during early embryonic development, including E2F6, remained to be addressed. In the submitted manuscript, using in vivo samples (E3.5 & E8.5) as well as ESCs and MEFs, Dahlet et al. demonstrate that E2F6 is critical for the silencing of a subset of germline genes, in part through recruitment of PRC1.6, H3K9me3 and the subsequent deposition of CpG methylation. More importantly, the authors show that E2F6 is required to initiate the silencing of germline genes in early embryonic stages but is dispensable for the maintenance in later/differentiated stages, indicating that this silencing pathway is essential for the establishment but not maintenance of DNAm. The authors' findings are novel, and most of the data are well presented. However, there are a few issues, discussed in detail below, which the authors should address before publication.

Major Points

Supplementary Figure 1E, shows a Venn diagram of the overlap between E2F6, MGA and L3MBTL2 ChIP-seq peaks in mouse ES cells. While 907 genes are common to each, 1643 genes are enriched only for MGA and L3MBTL2. Are there germline genes among these apparent PRC1.6 targets? Mentioning this and adding an example if so to panel F would be nice (only examples of the first category are currently shown). Indicates that there may be a subset of targets of MGA/MAX and PRC1.6 that are not bound by E2F6, which I believe is consistent with expression data from later figures.

In Supplementary Figure 2F, the authors show interestingly that few genes up in G9a^{-/-} E8.5 embryos are also up in E2F6^{-/-} embryos. Would nevertheless be informative for the authors to mention how many of the genes up in G9a^{-/-} embryos were identified as bound by E2F6 and/or MGA/MAX in ESCs. In other words, are some of the germline genes up in G9a^{-/-} embryos bound by E2F6 but simply maintained in a silent state by alternative silencing machinery?

On a related note, is there an overlap between germline genes upregulated in G9a or GLP KO ESCs (mining published data) and those upregulated in E2F6 KO ESCs? Considering gene lists from only E8.5 may be misleading, as DNAm may be less important at the earlier stage represented by ESCs in 2i, as pointed out by the authors.

On Page 5 & Figure 3: To address the question of whether E2F6 and MGA/MAX cooperate in the silencing of germline genes, the authors mention the presence of E2F6 and E-box motifs at around the TSSs of select germline genes. However, the binding of MGA is observed at the promoters of genes with no E-box (CACGTG) motif (such as Tuba3a, Gpat2, and Meioc) in Supplementary Figure 1F. Therefore, the authors should distinguish genes based on the observed binding of E2F6/MGA from their own and/or public ChIPseq data, rather than based just on the presence of motifs, and then should proceed with the relevant subsequent analyses/discussion. In Figure 3C, for example, why is a higher level of de-repression by Mga-KD in E2f6-KO ESCs not observed for Tuba3a, which shows bona fide MGA binding at its promoter? Perhaps MGA binding based on ChIPseq data is less predictive of

MGA function independent of E2F6 than the presence of an E-box motif in the promoter?

Might this also explain why, as shown in Figure 3D, RYBP and PCGF6 are reduced in E2f6-KO ESCs at the promoters of *Tuba3a*, *Gpat2*, and *Meioc*, despite the presence of MGA binding at these promoters?

On page 7, the authors state: "To this end, we first measured the expression of 14 E2F6 target genes in naïve E2f6+/+ and E2f6-/- ES cells grown in 2i conditions, which show global DNA hypomethylation similarly to blastocysts 39." The implication being that the germline genes of interest lack DNAm in 2i ESCs. However, as some genomic regions are resistant to loss of DNAm even under these conditions, the authors must show this, using for example published WGBS data from 2i ESCs. Are the germline genes indeed hypomethylated in this context?

Similarly, the authors state on page 7: "To corroborate this finding in vivo, we measured the expression of the E2F6 target genes *Tuba3a* and *Hormad1* in E2f6+/+ and E2f6-/- blastocysts and confirmed that E2F6 represses its target genes at this developmental stage (Fig 5B). This suggests that E2F6 initially represses germline genes by mechanisms other than DNA methylation in early embryonic cells before DNA methylation acquisition." However, again, the authors do not actually show the DNAm status of the genes of interest in blastocysts. Mining of published WGBS data should be carried out to confirm the conjecture that these gene promoters are NOT DNA methylated at this earlier stage of embryonic development.

Figure 5H & 5I: The observation that E2F6 does not directly interact with DNMT3B contradicts a previous report (Velasco et al. 2010, PNAS). The authors should discuss this point in more detail. Given that DNA methylation at the *Tuba3a* promoter is deposited in an E2F6-specific dimerization-dependent manner, it would be worth checking whether E2F6 recruits DNMT3B through heterodimerization with its partner protein DP1.

Minor points:

In the intro, where the authors mention the role of H3K4 trimethylation in inhibition of binding to H3 histone tails and allosteric activation of de novo DNA methyltransferases, would be nice if the recent papers (in NSMB, Gretarsson et al and Eckersley-Maslin et al) describing the role of *Dppa2* and *Dppa4* in this process were cited.

In this sentence in the intro. "Interestingly MAX, PCGF6, L3MBTL2 1 and RYBP have been linked to the repression of germline genes in mouse ES cells 23,29-32." MGA should be included and Stielow et al. (currently reference 37) cited.

Figure 1F: Gene names should be added to the figure.

Supplementary Tables 3 & 4: The authors should show lists of the downregulated genes in addition to the upregulated genes.

Supplementary Figure 2D: The authors should comment on why the 151 genes are repressed without E2F6 binding in WT ESCs.

Regarding Figure 2F, if possible, the authors should include ChIP-qPCR assay (like Figure 2C) from E8.5 embryos to test the binding of E2F6 in the promoters of E2F6 target genes.

Since the authors recently found that 137 germline genes acquire dense promoter CpG methylation in WT embryos and are derepressed in *Dnmt3a3b* double KO embryos at E8.5 (Dahlet et al. 2020, Nat Commun), it would be informative to show the overlap between those genes and genes upregulated in E2f6 KO E8.5 embryos, particularly as they do show that DNMT3B plays a role in DNAm of the loci of interest in Figure 5.

Regarding Figure 5A & Supplementary Figure 4, would be nice if the authors included E2F6 ChIP-qPCR for ESCs in 2i to confirm E2F6 binding to the promoters of target genes under these conditions.

The authors state on page 6: " Thus, repression of germline genes by E2F6 requires the DNA binding and marked box domains of E2F6." However, this is only demonstrated for 1 gene- Tuba3a. The sentence should be reworded.

The authors conclude on page 8 that: "Altogether, these results demonstrate that E2F6 is indispensable for recruiting DNA methylation-dependent silencing to its target genes during embryonic development." This does not seem the best way to summarize what they have shown here. Something more like: "Altogether, these results demonstrate that while E2F6 is indispensable for promoting DNA methylation-dependent silencing to its target genes during embryonic development, such DNA methylation is apparently not due to direct recruitment of DNMT3B by E2F6.

Last sentence of the first paragraph of the Discussion suffers from the same problem. The authors have interestingly shown here that this may not be due to direct "recruitment" of the de novo DNA methylation machinery. They should make this point clearly here, which they do expand upon later in the Discussion.

Regarding Figure 6C, ChIP-qPCR analysis of MEFs to determine whether E2F6 even binds to the promoters of select E2F6 target genes should be included.

Also, the authors should at least comment in the MEF section whether DNMT3B is expressed in these cells. Is failure of de novo DNAm in these cells simply due to the absence of this de novo DNMT, with only maintenance DNAm actually functioning?

In many of the figures, particularly when multiple different cell types/stages (ie ESCs vs blastocyst vs E8.5) are shown in the same figure, the authors should do a better job with labelling the panels to make it clear which time point is being analyzed.

Reviewer #2 (Remarks to the Author):

In this manuscript, Dahlet et al report a role for the E-box binding factor E2F6 in repressing transcription of a set of germline genes during early embryogenesis in the mouse. A distinguishing feature of these genes is that their promoter CpG islands become constitutively DNA methylated in somatic lineages. Prior work, mostly in mouse ES cells, had implicated E2F6. This study addresses the specificity and mechanism of the programmed DNA methylation induced by E2F6, including data from in vivo.

In brief, the authors show that:

1. E2F6 binds ~2000 sites in the mouse ES cell genome, predominantly at CpG islands, and with substantial overlap with components of the PRC1.6 complex;
2. in E2F6 deficient ES cells and in vivo, a subset of germline genes become de-repressed;
3. there is an overlap in the genes up-regulated by the absence of E2F6 and by absence of MGA or MAX, DNA-binding components of PRC1.6;
4. however, E2F6 is not required for PRC1.6 recruitment, indicating that part of E2F6 function at these genes is independent of PRC1.6;
5. E2F6 function depends not only on its DNA-binding domain, but also on the 'marked box domain', known to be a mediator of target-site selectivity of E2F proteins;
6. in vivo, E2F6 represses germline genes prior to de novo methylation, it is required for methylation of these genes, but becomes dispensable once long-term silencing is attained;
7. E2F6 antagonises binding of the activating E-box factor E2F1.

The study is a typically thorough analysis by the lab of Michael Weber, the experiments are done to a very high standard, with validation often at multiple levels, thereby providing compelling results.

Questions of mechanism remain, in particular, the nature of the repression by E2F6 independent of PRC1.6, and what are the properties of the E2F6-regulated promoters by which they are preferentially bound by E2F6 rather than E2F1. In relation to the latter, are the authors able to provide information or informed speculation; for example, anything about the conformation of E2F6 binding motifs of the germline genes that could suggest why E2F6 could out-compete E2F1 binding at these targets? Further analysis of sequence composition, other motifs? The authors speculate that the preference for E2F6 binding over E2F1 is via protein-protein interactions, suggesting the dependence on additional, specifying factors whose binding might be revealed from the promoter sequences.

In order to demonstrate that E2F6 is only transiently required for germline gene repression and DNA methylation, the authors compare MEFs from E2F6-knock-out embryos, which constitutively lack E2F6, with MEFs from wild-type embryos in which E2f6 is subsequently inactivated by CRISPR/Cas9 deletion. This is an acceptable approach, but conditional or tissue-specific ablation would be a more elegant or definitive demonstration of the temporal dependence of E2F6.

Minor comments

There is too much interpretation in a few of the figure legends: the legends should just give the necessary information to explain what the components of the figures represents, not an interpretation.

Some figure panels are missing statistical treatments (e.g., Fig. 3E).

Reviewer #3 (Remarks to the Author):

In this manuscript, Dahlet et al. reported that E2F6-containing ncPRC1.6 complex, preferentially bound to non-methylated CGIs and hypermethylated germline-specific gene promoters in mESCs. However, genetic deletion of E2F6 in mESCs or in early embryos only de-repressed limited germline-specific genes. The function of E2F6 in repressing germline genes depended on its DBD and MB domains, and its deletion did not affect the recruitment of other ncPRC1.6 subunits to the gene promoters. In addition, the authors found that E2F6 competed with E2F1 and predominantly bound to the repressed germline gene promoters. Deletion of E2F6 caused DNA hypomethylation of germline-specific gene promoters in mESCs and MEF cells isolated from early embryos, suggesting that E2F6 silences the germline-specific genes by inducing DNA hypermethylation at their promoters during early developmental stages.

Main points:

1. The authors showed that ncPRC1.6 subunits co-occupied the E2F6 binding sites at most non-methylated CGIs (Fig. 1), as well as the germline-specific gene promoters that are hypermethylated (Fig.1F). To strengthen their observations, the authors should: (1) provide binding profiles of all ncPRC1.6 subunits including the catalytic subunits Ring1a/Ring1b at germline-specific gene promoters in the E2F6-null cells to confirm the ncPRC1.6 occupancy at germline-specific gene promoters is E2F6 independent; (2) examine whether the ncPRC1.6 complex is still intact in the E2F6-null cells; (3) examine the DNA methylation states to confirm DNA hypermethylation of promoters of germline-specific gene promoters.

2. According to the model, E2F6 binds to the unmethylated germline-specific gene promoters and induces DNA hypermethylation at promoters (Fig. 5H). However, it is hard to explain that most CGIs bound by E2F6 remain unmethylated (Fig. 1). The author should further examine the difference between these two group of E2F6 bound sites, for example, the DNA sequence features, chromatin accessibility, and other histone modifications such as H3K4me3, to figure out the underlying mechanisms leading to the difference in DNA methylation between these two groups of E2F6 bound

regions.

3. It is interesting that although E2F6 binds to more than two thousand CGIs, only very limited germline-specific genes are de-repressed after loss of E2F6. In addition, Fig. 3E showed Ring1a/1b-dko did not de-repress germline-specific genes, suggesting the E2F6-mediated germline-specific gene silencing is PRC1-independent. It has been reported ncPRC1.6 interacts with multiple factors such as G9a and HDACs to silence gene expression (cell stem cell 2012 11: 319032), the authors should perform in-depth analysis along these direction to figure out the mechanisms by which E2F6 silences the germline-specific genes.

4. Figure 5A-B. The authors should perform bisulfite sequencing analysis to confirm that E2F6-targeted germline-specific gene promoters have DNA hypomethylation in 2i culture condition or in early blastocysts before they can draw the conclusion that the E2F6-mediated gene repression is independent of DNA methylation at early developmental stages.

5. Figure 5 shows the loss of E2F6 leads to hypomethylation at germline-specific gene promoters and E2F6 does not recruit DNMT3b. The authors should test the other alternative hypotheses, for example, whether E2F6 recruits DNMT1 or Uhrf1 to maintain DNA hypermethylation at the germline-specific gene promoters.

Minor points:

1. Figure 1F: The authors should annotate the reference genes in the genome browser.

In summary, although this study is addressing an interesting question, the main caveat is that the study did not provide new insights or conclusive answers to the fundamental questions that the authors intend to address: (1) how the E2F6-containing ncPRC1.6 is recruited to germline-specific promoters and induces DNA hypermethylation specifically at these promoters? (2) what are the molecular mechanisms by which E2F6 specifically represses germline-specific gene expression? Therefore, I do not recommend publish this study at Nature Communications.

Reviewer #1 (Remarks to the Author):

The expression of germline genes involved in meiosis and gametogenesis is repressed by multiple mechanisms in early embryonic and somatic cells. For example, the non-canonical PRC1.6 complex, the histone methyltransferases G9A/GLP, as well as SETDB1, repress both unique and overlapping sets of germline genes in pluripotent mouse ESCs through the accumulation of H2AK119ub1 and H3K9me2/3, respectively at their promoter regions (Ehdoh et al. 2017, eLife; Maeda et al. 2013, Nat Commun; Karimi et al. 2011, Cell Stem Cell), guided in part by the DNA binding factors MGA/MAX. In mouse somatic cells, the transcription factor E2F6 was reported to have a related role in the silencing of such germline genes via the recruitment of the DNA methyltransferase DNMT3B, followed by the deposition of CpG methylation (Velasco et al. 2010, PNAS). However, these findings were predominantly derived from somatic cells of a late embryonic stage, or for other PRC1.6 subunits, ESCs. Therefore, the “order of events” and function(s) of each chromatin factor/complex during early embryonic development, including E2F6, remained to be addressed. In the submitted manuscript, using in vivo samples (E3.5 & E8.5) as well as ESCs and MEFs, Dahlet et al. demonstrate that E2F6 is critical for the silencing of a subset of germline genes, in part through recruitment of PRC1.6, H3K9me3 and the subsequent deposition of CpG methylation. More importantly, the authors show that E2F6 is required to initiate the silencing of germline genes in early embryonic stages but is dispensable for the maintenance in later/differentiated stages, indicating that this silencing pathway is essential for the establishment but not maintenance of DNAm. The authors' findings are novel, and most of the data are well presented. However, there are a few issues, discussed in detail below, which the authors should address before publication.

We thank the reviewer for these positive comments.

Major Points

Supplementary Figure 1E, shows a Venn diagram of the overlap between E2F6, MGA and L3MBTL2 ChIP-seq peaks in mouse ES cells. While 907 genes are common to each, 1643 genes are enriched only for MGA and L3MBTL2. Are there germline genes among these apparent PRC1.6 targets? Mentioning this and adding an example if so to panel F would be nice (only examples of the first category are currently shown). Indicates that there may be a subset of targets of MGA/MAX and PRC1.6 that are not bound by E2F6, which I believe is consistent with expression data from later figures.

We agree that this is an interesting point. We investigated this and found several germline genes bound and repressed by MGA but not E2F6. We added an example of germline gene bound by MGA and PRC1.6 but not E2F6 in the **Supplementary Figure 3b**, which correlates with expression data from the **Figure 3b**. We modified the text to highlight this point with the following sentence:

*"MGA also represses additional germline genes that are not repressed by E2F6, such as *Tex101*, *Tdrd9*, *Taf7l*, *Mov10l1* or *Tex19.2* (Supplementary Fig 3b and Fig 3b)"*

In Supplementary Figure 2F, the authors show interestingly that few genes up in G9a^{-/-} E8.5 embryos are also up in E2F6^{-/-} embryos. Would nevertheless be informative for the authors to mention how many of the genes up in G9a^{-/-} embryos were identified as bound by E2F6 and/or MGA/MAX in ESCs. In other words, are some of the germline genes up in G9a^{-/-} embryos bound by E2F6 but simply maintained in a silent state by alternative silencing machinery?

We thank the reviewer for this comment and investigated this point. Interestingly, most germline genes up in G9a^{-/-} embryos were not identified as bound by E2F6, suggesting that G9a and E2F6 act in distinct pathways. We added this sentence in the text **page 8** to highlight this information:

"In addition, most germline genes repressed by G9a were not identified as bound by E2F6, suggesting that they act in distinct pathways".

On a related note, is there an overlap between germline genes upregulated in G9a or GLP KO ESCs (mining published data) and those upregulated in E2F6 KO ESCs? Considering gene lists from only E8.5 may be misleading, as DNAm may be less important at the earlier stage represented by ESCs in 2i, as pointed out by the authors.

We thank the reviewer for this interesting comment. We compared the expression of germline genes in *E2f6* KO and *G9a* KO ESCs (Mozzetta et al.) and found that E2F6-target genes are not significantly derepressed in *G9a* KO ESCs (**Supplementary Figure 6a-b**), confirming the data from E8.5 embryos. Furthermore we now include an analysis of E2F6 interaction partners by mass-spectrometry (see below) and added a sentence in the text to emphasize that G9a and GLP are not enriched in the E2F6 interactome. Taken together these data reinforce our initial result that E2F6 function is independent of G9a in ES cells and embryos.

On Page 5 & Figure 3: To address the question of whether E2F6 and MGA/MAX cooperate in the silencing of germline genes, the authors mention the presence of E2F6 and E-box motifs at around the TSSs of select germline genes. However, the binding of MGA is observed at the promoters of genes with no E-box (CACGTG) motif (such as *Tuba3a*, *Gpat2*, and *Meioc*) in Supplementary Figure 1F. Therefore, the authors should distinguish genes based on the observed binding of E2F6/MGA from their own and/or public ChIPseq data, rather than based just on the presence of motifs, and then should proceed with the relevant subsequent analyses/discussion. In Figure 3C, for example, why is a higher level of derepression by Mga-KD in E2f6-KO ESCs not observed for *Tuba3a*, which shows bona fide MGA binding at its promoter? Perhaps MGA binding based on ChIPseq data is less predictive of MGA function independent of E2F6 than the presence of an E-box motif in the promoter?

It is well known that ChIP peaks may result from indirect recruitment because of interaction with another protein that in turn binds DNA. Because of the strong E2F6-MGA interaction, some MGA peaks are probably indirect. Indeed, it has been shown that MGA is able to bind a subset of loci independent of its DNA-binding activity, probably owing to indirect recruitment (Stielow et al., PLoS Genet 2018). Using the presence of DNA motifs is a widely used approach to differentiate direct from indirect binding and function of TFs. We thus believe that the presence of peaks + motifs is more predictive of E2F6/MGA function than peaks alone and therefore prefer to distinguish genes based on these criteria.

Might this also explain why, as shown in Figure 3D, RYBP and PCGF6 are reduced in E2f6-KO ESCs at the promoters of *Tuba3a*, *Gpat2*, and *Meioc*, despite the presence of MGA binding at these promoters?
Yes, see above.

On page 7, the authors state: “To this end, we first measured the expression of 14 E2F6 target genes in naïve E2f6^{+/+} and E2f6^{-/-} ES cells grown in 2i conditions, which show global DNA hypomethylation similarly to blastocysts 39.” The implication being that the germline genes of interest lack DNAm in 2i ESCs. However, as some genomic regions are resistant to loss of DNAm even under these conditions, the authors must show this, using for example published WGBS data from 2i ESCs. Are the germline genes indeed hypomethylated in this context?

We thank the reviewer for raising this important point. We confirmed by bisulfite sequencing analysis that the promoters of four E2F6-target genes from Figure 5b (*Gpat2*, *Tuba3a*, *Hormad1*, *Meioc*) are strongly hypomethylated in our ES cells grown in 2i conditions compared to serum (**Supplementary Figure 5b**). Furthermore, we also performed RNAi mediated knockdown of *E2f6* in *Dnmt*-TKO ESCs and showed that this increases the expression of germline genes with a similar magnitude than in WT ESCs (**Supplementary Figure 5e**). These data reinforce the conclusion that E2F6 represses germline genes by DNA-methylation independent mechanisms in early embryonic cells.

Similarly, the authors state on page 7: “To corroborate this finding in vivo, we measured the expression of the E2F6 target genes *Tuba3a* and *Hormad1* in E2f6^{+/+} and E2f6^{-/-} blastocysts and confirmed that E2F6 represses its target genes at this developmental stage (Fig 5B). This suggests that E2F6 initially represses germline genes by mechanisms other than DNA methylation in early embryonic cells before DNA methylation acquisition.” However, again, the authors do not actually show the DNAm status of the genes of interest in blastocysts. Mining of published WGBS data should be carried out to confirm the conjecture that these gene promoters are NOT DNA methylated at this earlier stage of embryonic development.

We thank again the reviewer for raising this important point. As requested, we included WGBS data in E3.5 blastocysts from the same genetic background (C57BL/6J) showing that the *Tuba3a* and *Hormad1* promoters are hypomethylated at this developmental stage (**Supplementary Figure 5a**).

Figure 5H & 5I: The observation that E2F6 does not directly interact with DNMT3B contradicts a previous report (Velasco et al. 2010, PNAS). The authors should discuss this point in more detail. Given that DNA methylation at the *Tuba3a* promoter is deposited in an E2F6-specific dimerization-dependent manner, it would be worth checking whether E2F6 recruits DNMT3B through heterodimerization with its partner protein DP1.

We agree that, in light of the literature, the E2F6-DNMT3B interaction deserved further investigation. We confirmed our negative co-IP results using different experimental conditions and two different DNMT3B antibodies (data not shown). Furthermore, we analyzed the E2F6 interactome by Mass-spectrometry. While we observed a strong association with PRC1.6 (**Figure 1i**), we found DNMT3B only slightly enriched in E2F6-bound proteins below the threshold of significance (**Supplementary Figure 8**), arguing against a strong direct interaction. We included this result in the revised manuscript (**page 9**). Unfortunately we were unable to optimize experimental conditions to investigate the DP1-DNMT3B interaction. In support of our results, DNMT3B was never found in PRC1.6 complexes in previous publications. Furthermore, the Velasco experiment was performed by strong overexpression of E2F6 and DNMT3B in human HEK293 cells, which might not recapitulate physiological conditions from mouse embryonic cells. We now expand on this point in the revised discussion.

Minor points:

In the intro, where the authors mention the role of H3K4 trimethylation in inhibition of binding to H3 histone tails and allosteric activation of de novo DNA methyltransferases, would be nice if the recent papers (in NSMB, Gretarsson et al and Eckersley-Maslin et al) describing the role of *Dppa2* and *Dppa4* in this process were cited.

We thank the reviewer for this suggestion and added these two references in the introduction.

In this sentence in the intro. “Interestingly MAX, PCGF6, L3MBTL2 1 and RYBP have been linked to the repression of germline genes in mouse ES cells 23,29-32.” MGA should be included and Stielow et al. (currently reference 37) cited.

As requested, we added MGA and the Stielow et al. reference.

Figure 1F: Gene names should be added to the figure.

We added the gene names in the **Figure 1f**.

Supplementary Tables 3 & 4: The authors should show lists of the downregulated genes in addition to the upregulated genes.

We now indicate the downregulated genes in addition to the upregulated genes in the Supplementary Tables.

Supplementary Figure 2D: The authors should comment on why the 151 genes are repressed without E2F6 binding in WT ESCs.

We added the following sentence in the text:

"Genes upregulated in E2f6^{-/-} cells without E2F6 binding probably reflect indirect or clonal effects."

Regarding Figure 2F, if possible, the authors should include ChIP-qPCR assay (like Figure 2C) from E8.5 embryos to test the binding of E2F6 in the promoters of E2F6 target genes.

We thank the reviewer for this very interesting suggestion, however we were unable to adapt the E2F6 ChIP protocol to very low amount of cells applicable to E8.5 embryos.

Since the authors recently found that 137 germline genes acquire dense promoter CpG methylation in WT embryos and are derepressed in Dnmt3a3b double KO embryos at E8.5 (Dahlet et al. 2020, Nat Commun), it would be informative to show the overlap between those genes and genes upregulated in E2f6 KO E8.5 embryos, particularly as they do show that DNMT3B plays a role in DNAm of the loci of interest in Figure 5.

We thank the reviewer for this suggestion. We now show that there is a strong overlap between the genes upregulated in DKO and E2f6 KO embryos (**Supplementary Figure 7g**) and added the following sentence in the text:

"Furthermore, most germline genes repressed by E2F6 are derepressed in Dnmt3a/b double-knockout embryos lacking de novo methylation (ref 15) (Supplementary Fig 7g, $p=5.28e-26$, hypergeometric test), demonstrating the importance of DNA methylation for E2F6-induced repression in embryos."

Regarding Figure 5A & Supplementary Figure 4, would be nice if the authors included E2F6 ChIP-qPCR for ESCs in 2i to confirm E2F6 binding to the promoters of target genes under these conditions.

While this manuscript was under revision, an E2F6 ChIP-seq from 2i ESCs was published (PMID 32822591). To address the reviewer's point, we now include a comparison of E2F6 ChIP-seq in serum and 2i ESCs, which shows that E2F6 binding to germline gene promoters is stable under both conditions (**Supplementary Figure 5c-d**).

The authors state on page 6: "Thus, repression of germline genes by E2F6 requires the DNA binding and marked box domains of E2F6." However, this is only demonstrated for 1 gene- Tuba3a. The sentence should be reworded.

We reworded the sentence as follows: *"Thus, repression of the Tuba3a gene by E2F6 requires the DNA binding and marked box domains of E2F6."*

Furthermore we extended the results with the marked box mutant E2F6-MB4 to four other germline genes in **Figure 4f**.

The authors conclude on page 8 that: "Altogether, these results demonstrate that E2F6 is indispensable for recruiting DNA methylation-dependent silencing to its target genes during embryonic development." This does not seem the best way to summarize what they have shown here. Something more like: "Altogether, these results demonstrate that while E2F6 is indispensable for promoting DNA methylation-dependent silencing to its target genes during embryonic development, such DNA methylation is apparently not due to direct recruitment of DNMT3B by E2F6.

Last sentence of the first paragraph of the Discussion suffers from the same problem. The authors have interestingly shown here that this may not be due to direct "recruitment" of the de novo DNA methylation machinery. They should make this point clearly here, which they do expand upon later in the Discussion.

We thank the reviewer for these suggestions and modified the sentences as follows:

Page 9: "Altogether, these results demonstrate that E2F6 is indispensable for promoting DNA methylation-dependent silencing to its target genes during embryonic development, apparently through indirect recruitment of DNMT3B."

Discussion: "E2F6 is critical to promote DNA methylation and initiate lifelong epigenetic silencing of a subset of germline genes during mouse development."

We also modified the discussion to discuss in more details that our data support an indirect rather than direct recruitment mechanism.

Regarding Figure 6C, ChIP-qPCR analysis of MEFs to determine whether E2F6 even binds to the promoters of select E2F6 target genes should be included.

It has been shown previously by ChIP that E2F6 binds to the promoters of germline genes in MEFs (Velasco *et al.*, PNAS 2010). We added a sentence in the text **page 10** to highlight this point.

Also, the authors should at least comment in the MEF section whether DNMT3B is expressed in these cells. Is failure of *de novo* DNAm in these cells simply due to the absence of this *de novo* DNMT, with only maintenance DNAm actually functioning?

We agree that this is an important point. We performed western blot to show that there is no detectable expression of DNMT3B in MEFs (**Supplementary Figure 9h**) and added a sentence in the text to highlight that incapacity to *de novo* methylate E2F6 target genes in MEFs might be due to the near absence of DNMT3B in these cells.

In many of the figures, particularly when multiple different cell types/stages (ie ESCs vs blastocyst vs E8.5) are shown in the same figure, the authors should do a better job with labelling the panels to make it clear which time point is being analyzed.

We thank the reviewer for this criticism, which helps improve the clarity of the figures. We improved the labelling of the panels in the **Figures 2, 5 and Supplementary Figure 7**.

Reviewer #2 (Remarks to the Author):

In this manuscript, Dahlet et al report a role for the E-box binding factor E2F6 in repressing transcription of a set of germline genes during early embryogenesis in the mouse. A distinguishing feature of these genes is that their promoter CpG islands become constitutively DNA methylated in somatic lineages. Prior work, mostly in mouse ES cells, had implicated E2F6. This study addresses the specificity and mechanism of the programmed DNA methylation induced by E2F6, including data from *in vivo*.

In brief, the authors show that:

1. E2F6 binds ~2000 sites in the mouse ES cell genome, predominantly at CpG islands, and with substantial overlap with components of the PRC1.6 complex;
2. in E2F6 deficient ES cells and *in vivo*, a subset of germline genes become de-repressed;
3. there is an overlap in the genes up-regulated by the absence of E2F6 and by absence of MGA or MAX, DNA-binding components of PRC1.6;
4. however, E2F6 is not required for PRC1.6 recruitment, indicating that part of E2F6 function at these genes is independent of PRC1.6;
5. E2F6 function depends not only on its DNA-binding domain, but also on the 'marked box domain', known to be a mediator of target-site selectivity of E2F proteins;

6. in vivo, E2F6 represses germline genes prior to de novo methylation, it is required for methylation of these genes, but becomes dispensable once long-term silencing is attained;
7. E2F6 antagonises binding of the activating E-box factor E2F1.

The study is a typically thorough analysis by the lab of Michael Weber, the experiments are done to a very high standard, with validation often at multiple levels, thereby providing compelling results.

We thank the reviewer for these positive comments.

Questions of mechanism remain, in particular, the nature of the repression by E2F6 independent of PRC1.6, and what are the properties of the E2F6-regulated promoters by which they are preferentially bound by E2F6 rather than E2F1. In relation to the latter, are the authors able to provide information or informed speculation; for example, anything about the conformation of E2F6 binding motifs of the germline genes that could suggest why E2F6 could out-compete E2F1 binding at these targets? Further analysis of sequence composition, other motifs? The authors speculate that the preference for E2F6 binding over E2F1 is via protein-protein interactions, suggesting the dependence on additional, specifying factors whose binding might be revealed from the promoter sequences.

We agree that the mechanisms of E2F6-mediated repression and the properties of germline genes that make them sensitive to E2F6-mediated repression remained elusive in the reviewed version of the manuscript.

The observed specificity could be mediated by the DNA sequence or DNA shape. We did not find any enrichment of DNA sequence feature or binding motif in germline genes that could reveal candidate specifying factors (not shown). Furthermore, inspection of the predicted DNA structures of E2F6 binding sites and their flanking regions did not reveal structural differences in the recognition motifs. Moreover, as we demonstrate in the **Figure 4f**, replacing the E2F6-DBD (DNA binding domain) with the DBD of E2F1 does not impair efficient repression of E2F6 target genes.

Instead we show that the specificity of E2F6-mediated repression is mediated by the marked box (MB) domain. This has already been documented for other E2F family members. Domain swapping experiments revealed that chimeric E2F1 and E2F3 proteins generate expression signatures that reflect the origin of the MB (Black et al., Proc Natl Acad Sci USA 2005). In the first version of the manuscript, we showed that the exchange of the E2F6 MB with the E2F4 MB results in inefficient repression of the Tuba3a gene. To demonstrate that this is not restricted to Tuba3a, we extended this analysis to 5 germline genes in the new **Figure 4f** and show that the E2F4 MB is unable to act as functional substitute for target gene repression at all tested genes. We speculate in the discussion that the MB could mediate specific local protein-protein interactions at the sites of germline promoters.

To explore the mechanisms of E2F6-mediated repression of germline genes independent of DNA methylation and PRC1.6, we searched for chromatin features that distinguish germline genes repressed by E2F6 from the other genes bound but not repressed by E2F6. We found that germline genes are specifically enriched for H3K9me3 and SETDB1, and we now provide new data reinforcing the model that E2F6 represses germline genes in part by favoring H3K9me3 deposition: (1) E2F6-repressed genes are enriched for SETDB1 binding and H3K9me3 compared to genes bound but not repressed by E2F6 (**Supplementary Figure 6e**), (2) E2F6 target genes are derepressed in Setdb1-KO ES cells (**Supplementary Figure 6a-b**), (3) H3K9me3 is reduced at E2F6-target genes in E2f6 KO ES cells as already shown in the first version of the manuscript (**Supplementary Figure 6f**).

In order to demonstrate that E2F6 is only transiently required for germline gene repression and DNA methylation, the authors compare MEFs from E2f6-knock-out embryos, which constitutively lack E2F6,

with MEFs from wild-type embryos in which E2f6 is subsequently inactivated by CRISPR/Cas9 deletion. This is an acceptable approach, but conditional or tissue-specific ablation would be a more elegant or definitive demonstration of the temporal dependence of E2F6.

We agree that the conditional inactivation of *E2f6* would be an elegant approach, however we were unable to perform this experiment because of the lack of available *E2f6*-2Lox mice. As an alternative, we now show that the E2F6 protein becomes barely detectable in several organs such as liver or kidney after birth (4 weeks of age) and that the absence of detectable E2F6 protein does not hinder the silencing of germline genes in these organs (**Supplementary Figure 9f-g**). These new data reinforce the model that E2F6 is dispensable for the maintenance of epigenetic silencing in differentiated organs.

Minor comments

There is too much interpretation in a few of the figure legends: the legends should just give the necessary information to explain what the components of the figures represents, not an interpretation. We apologize for that and adapted the legends of **Figures 2, 3 and 4** to remove data interpretations.

Some figure panels are missing statistical treatments (e.g., Fig. 3E).

We added statistical treatment to **Figures 2e, 3e, 5a, 5i** and **Supplementary Figures 4b, 6a, 6c, 6f, 7g**.

Reviewer #3 (Remarks to the Author):

In this manuscript, Dahlet et al. reported that E2F6-containing ncPRC1.6 complex, preferentially bound to non-methylated CGIs and hypermethylated germline-specific gene promoters in mESCs. However, genetic deletion of E2F6 in mESCs or in early embryos only de-repressed limited germline-specific genes. The function of E2F6 in repressing germline genes depended on its DBD and MB domains, and its deletion did not affect the recruitment of other ncPRC1.6 subunits to the gene promoters. In addition, the authors found that E2F6 competed with E2F1 and predominantly bound to the repressed germline gene promoters. Deletion of E2F6 caused DNA hypomethylation of germline-specific gene promoters in mESCs and MEF cells isolated from early embryos, suggesting that E2F6 silences the germline-specific genes by inducing DNA hypermethylation at their promoters during early developmental stages.

Main points:

1. The authors showed that ncPRC1.6 subunits co-occupied the E2F6 binding sites at most non-methylated CGIs (Fig. 1), as well as the germline-specific gene promoters that are hypermethylated (Fig.1F). To strengthen their observations, the authors should: (1) provide binding profiles of all ncPRC1.6 subunits including the catalytic subunits Ring1a/Ring1b at germline-specific gene promoters in the E2F6-null cells to confirm the ncPRC1.6 occupancy at germline-specific gene promoters is E2F6 independent; (2) examine whether the ncPRC1.6 complex is still intact in the E2F6-null cells; (3) examine the DNA methylation states to confirm DNA hypermethylation of promoters of germline-specific gene promoters.

We thank the reviewer for these valuable comments.

(1) Because of time constraints and difficulties during the COVID-19 period, we could not perform additional ChIPs for all PRC1.6 subunits. We would like to emphasize that the role of E2F6 as a targeting module of PRC1.6 has been addressed in two previous publications (Stielow et al., PLoS Genet 2018 ; Scelfo et al., Mol Cell 2019). In line with our data, both studies observed that E2F6 has a small contribution to PRC1.6 recruitment compared to MGA.

(2) To examine whether the PRC1.6 complex is intact in *E2f6*-null cells, we performed western blot analysis of all PRC1.6 proteins in WT and *E2f6*^{-/-} ES cells (**Supplementary Fig 4c**). While the loss of MGA leads to complex destabilization and reduced protein levels of several PRC1.6 proteins (Stielow et al., PLoS Genet 2018 ; Scelfo et al., Mol Cell 2019, Qin et al., Science Advances 2021), the protein levels of PRC1.6 proteins are unaffected in *E2f6*-null cells, strongly suggesting that the PRC1.6 complex is intact.

(3) DNA hypermethylation of promoters of germline genes has been demonstrated in many previous publications (for example refs 3 and 15) and is also confirmed in our WGBS and RRBS data in embryos and ES cells (**Figure 5d, g**).

2. According to the model, E2F6 binds to the unmethylated germline-specific gene promoters and induces DNA hypermethylation at promoters (Fig. 5H). However, it is hard to explain that most CGIs bound by E2F6 remain unmethylated (Fig. 1). The author should further examine the difference between these two group of E2F6 bound sites, for example, the DNA sequence features, chromatin accessibility, and other histone modifications such as H3K4me3, to figure out the underlying mechanisms leading to the difference in DNA methylation between these two groups of E2F6 bound regions.

We thank the reviewer for this very interesting suggestion. We performed an in depth analysis of DNA sequence and chromatin features to examine the differences between germline genes bound and repressed by E2F6 and the other genes bound but not repressed by E2F6. We were unable to find any difference in DNA sequence features and motifs (not shown). When we investigated chromatin features, we found that the main difference is the enrichment of H3K9me3/SETDB1 and depletion of H3K4me3 at germline genes compared to the genes bound but not repressed by E2F6 (**Supplementary Fig 6e**). Furthermore, we added new ChIP-qPCR data showing that E2F6 maintains low levels of H3K4me3 at germline genes (**Figure 5j**). These new data support the model that E2F6 induces DNA methylation of germline genes indirectly by its specific ability to repel activating TFs and H3K4me3 at these sites, which overrides the intrinsic protection of CGIs against DNA methylation.

3. It is interesting that although E2F6 binds to more than two thousand CGIs, only very limited germline-specific genes are de-repressed after loss of E2F6. In addition, Fig. 3E showed *Ring1a/1b*-dKO did not de-repress germline-specific genes, suggesting the E2F6-mediated germline-specific gene silencing is PRC1-independent. It has been reported ncPRC1.6 interacts with multiple factors such as G9a and HDACs to silence gene expression (cell stem cell 2012 11: 319032), the authors should perform in-depth analysis along these direction to figure out the mechanisms by which E2F6 silences the germline-specific genes. We agree with the reviewer's comments and further investigated the mechanisms of E2F6-mediated repression.

First, we characterized the E2F6 interactome by immunoprecipitation followed by mass-spectrometry (**Figure 1i**). While we confirmed a strong interaction between E2F6 and PRC1.6, the E2F6 interactome did not reveal an epigenetic partner that could explain E2F6-mediated repression by direct interaction. In particular, we found no evidence for an interaction with G9a or HDACs (**Supplementary Table 3**), in agreement with several other interactome studies that failed to detect G9a in PRC1.6 (refs 25-26-28). Furthermore, as also requested by the reviewer 1, we compared gene expression in *E2f6* KO and *G9a* KO ES cells and found that *G9a* does not repress *E2f6*-target genes (**Supplementary Figure 6a**). These data suggest that E2F6 function is independent of G9a.

Second, we searched for discriminative chromatin features between germline genes bound and repressed by E2F6 and the other genes bound but not repressed by E2F6, and we now provide new data reinforcing the model that E2F6 represses germline genes in part by favoring H3K9me3 deposition: (1)

E2F6-repressed genes are enriched for SETDB1 binding and H3K9me3 compared to genes bound but not repressed by E2F6 (**Supplementary Figure 6e**), (2) E2F6 target genes are derepressed in Setdb1-KO ES cells (**Supplementary Figure 6a-b**), (3) H3K9me3 is reduced at E2F6-target genes in E2f6 KO ES cells as already shown in the first version of the manuscript (**Supplementary Figure 6f**).

4. Figure 5A-B. The authors should perform bisulfite sequencing analysis to confirm that E2F6-targeted germline-specific gene promoters have DNA hypomethylation in 2i culture condition or in early blastocysts before they can draw the conclusion that the E2F6-mediated gene repression is independent of DNA methylation at early developmental stages.

We thank the reviewer for raising these important points. As also requested by the reviewer 1, we included WGBS data in E3.5 blastocysts from the same genetic background (C57BL6/J) showing that the *Tuba3a* and *Hormad1* promoters are hypomethylated at this developmental stage (**Supplementary Figure 5a**). Furthermore, we confirmed by bisulfite sequencing analysis that the promoters of four E2F6-target genes from Figure 5b (*Gpat2*, *Tuba3a*, *Hormad1*, *Meioc*) are strongly hypomethylated in our ES cells in 2i conditions compared to serum (**Supplementary Figure 5b**). Finally, we also performed RNAi mediated knockdown of *E2f6* in *Dnmt*-TKO ESCs and showed that this increases the expression of germline genes with a similar magnitude than in WT ESCs (**Supplementary Figure 5e**). Altogether, these data reinforce the conclusion that E2F6 represses germline genes by DNA-methylation independent mechanisms in early embryonic cells.

5. Figure 5 shows the loss of E2F6 leads to hypomethylation at germline-specific gene promoters and E2F6 does not recruit DNMT3b. The authors should test the other alternative hypotheses, for example, whether E2F6 recruits DNMT1 or Uhrf1 to maintain DNA hypermethylation at the germline-specific gene promoters.

We thank the reviewer for this suggestion. As described above, we characterized the E2F6 interactome by mass spectrometry and investigated the interactions between E2F6 and the DNA methylation machinery. We found no evidence for an interaction between E2F6 and DNMT3A, DNMT1 or UHRF1 (**Supplementary Figure 8**), suggesting that E2F6 does not promote DNA methylation by direct interaction with the DNA methylation machinery.

Minor points:

1. Figure 1F: The authors should annotate the reference genes in the genome browser.

We added the gene names in the **Figure 1f**.

REVIEWERS' COMMENTS

Reviewer #1 (Remarks to the Author):

The authors have addressed all of my concerns and edited the manuscript accordingly. Also added nice new data supporting their original model. Thus, the manuscript is much improved and worthy of publication in Nat Comm in my view. I have only two minor suggestions.

1. In response to my question: "Perhaps MGA binding based on ChIPseq data is less predictive of MGA function independent of E2F6 than the presence of an E-box motif in the promoter?"

The authors reply: "It is well known that ChIP peaks may result from indirect recruitment because of interaction with another protein that in turn binds DNA. Because of the strong E2F6-MGA interaction, some MGA peaks are probably indirect. Indeed, it has been shown that MGA is able to bind a subset of loci independent of its DNA-binding activity, probably owing to indirect recruitment (Stielow et al., PLoS Genet 2018). Using the presence of DNA motifs is a widely used approach to differentiate direct from indirect binding and function of TFs. We thus believe that the presence of peaks + motifs is more predictive of E2F6/MGA function than peaks alone and therefore prefer to distinguish genes based on these criteria."

Would be great if the authors mentioned this rationale/observation in the text. I think it is important information to point out for the non-expert reader, even if a widely applied approach.

Pg 7, line 31- "...consistent with..." not "...consistently with..."

Reviewer #3 (Remarks to the Author):

The authors have addressed my concerns in their revised manuscript. I do not have additional comments and recommend for publication in NC.

We thank the reviewers for their positive comments on the revised manuscript.

Reviewer #1 (Remarks to the Author):

The authors have addressed all of my concerns and edited the manuscript accordingly. Also added nice new data supporting their original model. Thus, the manuscript is much improved and worthy of publication in Nat Comm in my view. I have only two minor suggestions.

1. In response to my question: "Perhaps MGA binding based on ChIPseq data is less predictive of MGA function independent of E2F6 than the presence of an E-box motif in the promoter?"

The authors reply: "It is well known that ChIP peaks may result from indirect recruitment because of interaction with another protein that in turn binds DNA. Because of the strong E2F6-MGA interaction, some MGA peaks are probably indirect. Indeed, it has been shown that MGA is able to bind a subset of loci independent of its DNA-binding activity, probably owing to indirect recruitment (Stielow et al., PLoS Genet 2018). Using the presence of DNA motifs is a widely used approach to differentiate direct from indirect binding and function of TFs. We thus believe that the presence of peaks + motifs is more predictive of E2F6/MGA function than peaks alone and therefore prefer to distinguish genes based on these criteria."

Would be great if the authors mentioned this rationale/observation in the text. I think it is important information to point out for the non-expert reader, even if a widely applied approach.

We added a sentence in the text page 5 to mention that it is well known that ChIP peaks may result from indirect recruitment, which justifies that we searched for E-box motifs to distinguish direct from indirect MGA function.

Pg 7, line 31- "...consistent with..." not "...consistently with..."

Done

Reviewer #3 (Remarks to the Author):

The authors have addressed my concerns in their revised manuscript. I do not have additional comments and recommend for publication in NC.